# Real-world size of objects serves as an axis of object space

Taicheng Huang [1], Yiying Song [2] & Jia Liu [1]

Our mind can represent various objects from physical world in an abstract and complex high-dimensional object space, with axes encoding critical features to quickly and accurately recognize objects. Among object features identified in previous neurophysiological and fMRI studies that may serve as the axes, objects' real-world size is of particular interest because it provides not only visual information for broad conceptual distinctions between objects but also ecological information for objects' affordance. Here we use deep convolutional neural networks (DCNNs), which enable direct manipulation of visual experience and units' activation, to explore how objects' real-world size is extracted to construct the axis of object space. Like the human brain, the DCNNs pre-trained for object recognition also encode objects' size as an independent axis of the object space. Further, we find that the shape of objects, rather than retinal size, context, task demands or texture features, is critical to inferring objects' size for both DCNNs and humans. In short, with DCNNs as a brain-like model, our study devises a paradigm supplemental to conventional approaches to explore the structure of object space, which provides computational support for empirical observations on human perceptual and neural representations of objects.

[1] Department of Psychology and Tsinghua Laboratory of Brain & Intelligence, Tsinghua University, Beijing, China. [2] Beijing Key Laboratory of Applied Experimental Psychology, Faculty of Psychology, Beijing Normal University, Beijing, China. ✉email: songyiying@bnu.edu.cn; liujiathu@tsinghua.edu.cn

Objects are complicated, and humans have developed an excellent ability to recognize them quickly and accurately in natural environment. A possible underlying mechanism for such a feat is to extract object features to construct an object space whose axes carry critical information regarding aspects of object properties[1,2]. Thus, object recognition is considered a computational problem of finding multiple axes to build a simplified division surface to separate different objects represented by object features projected from these axes[3]. Previous neurophysiological and fMRI studies suggest that this hypothetical object space is implemented in human's ventral temporal cortex (VTC) and non-human primates' inferotemporal cortex with axes such as real-world size (big versus small)[4–8], animacy (animate versus inanimate)[4,5,7,9,10] and curvature (spiky versus stubby)[9,11,12].

Among these object features, objects' real-world size, which is encoded along the medial fusiform sulcus (MFS)[8] in the VTC[7] and medial temporal lobe (MTL)[4], is of particular interest, because it provides critical information not only to support broad conceptual distinctions between objects but also for us to understand objects' relation to the environment and to interact with objects. Accordingly, the functionality of objects' real-world size suggests at least three sources that may account for the development of size sensitivity in the VTC. The most evident source is that objects of different sizes likely have distinct mid-level perceptual properties (e.g., local corners, junctions, and contours) that are extracted at the early stages of visual processing, as in visual search the target object is detected faster when it differs in real-world size with the distractor objects[13] and the cortical region with size sensitivity is evoked by unrecognizable objects of different sizes that only preserve coarse form information[14].

On the other hand, the real-world size describes the scale of an object in natural environment, which implies the potential layout of the object[6]. For example, the size of whales suggests its co-occurrence with seas but not creeks. Therefore, the size sensitivity may derive from the context information that describes co-occurrence of multiple objects and their relations to the environment observed in daily life. Finally, the real-world size provides heuristic information for affordance[15], in which a specific real-world size of an object is associated with a set of specific actions, but not every action. Accordingly, the size sensitivity may be guided by action, which is potentially realized through differential connectivity with dorsal-manipulation versus medial-navigation networks[16]. Note that these three sources for the development of the size sensitivity are not mutually exclusive, because in daily life they are usually tightly intermingled and therefore hardly decoupled in conventional experiments to evaluate the contribution of each source independently.

Deep convolutional neural networks (DCNNs) provide a new computational framework to explore sources in the development of size sensitivity in biological systems. First, recent advance in DCNNs shows great potential for simulating human and primates' ventral visual pathway in object recognition[17–21], such as retinotopy[22], semantic structure[23], coding scheme[24] and face representation[25]. Besides, in DCNNs we can selectively deprive specific visual experience[26], precisely activate or deactivate units' responses[24], and smoothly adjust levels of task demands[23], which provides unprecedented flexibility of conducting experiments. More importantly, a set of DCNNs are specifically designated for perception (i.e., object recognition) without top-down semantic modulation or action-based task demands; therefore, we can examine to what degree the axis of real-world size is evident in the object space constructed in the DCNNs that only analyze image statistics of objects.

To do this, we first examined whether a typical DCNN designated for object recognition, AlexNet[27], encoded objects' real-world size as an axis of object space. Then, we examined the sources for the development of the size axis by systematically manipulating factors of retinal size, context, task demands, shape and texture of objects. Finally, we used fMRI to examine whether the factors identified in DCNNs also contributed to the representation of objects' real-world size in the human brain.

## Results

**DCNNs encoded objects' real-world size as an axis of object space.** We first evaluated whether AlexNet pre-trained for object recognition encoded the feature of objects' real-world size by examining the correspondence of AlexNet's responses and objects' size. An ideal observer was constructed as a baseline to represent the size relation among objects, which was used to measure how closely the representation of objects' size matched the ground truth (Fig. 1a, Supplementary Data 1). We found that the representational similarity matrix (RSM) of objects' sizes at Conv4 layer of the AlexNet was highly correlated with that of the ideal observer (Fig. 1b) ($r = 0.96$, $p < 0.001$), suggesting that Conv4 layer represented feature of objects' size. In addition, to examine the similarity between AlexNet's responses and human's subjective experience on objects' size, we also measured human's judgment on objects' size where participants were instructed to choose a larger object from object pairs randomly sampled from the same dataset. We found human's subjective experience of object size was highly similar to AlexNet's responses to it (Fig. 1c; $r = 0.94$, $p < 0.001$), suggesting at least a weak equivalence between DCNN and human in representing the size feature of objects. Furthermore, the similarity was not restricted to Conv4 layer; instead, all convolution layers, except Conv1 layer, showed sensitivity to objects' size (Fig S1), with Conv4 layer showing the highest correspondence. Similar results were observed in DCNNs with different architectures as well (Fig S2).

To examine whether the size feature served as an axis of object space, we used principal component analysis (PCA) on Conv4's responses to 50,000 objects from the ImageNet validation dataset to construct an object space with 50 orthogonal axes, and the number of axes was determined based on the amount of the response variance explained (i.e., > 90%). By iteratively removing Conv4's response variance aligned to an axis, we examined the decrease in correspondence between the RSM of the residual variance and that of the ideal observer. We found that only did the removal of the second PC (PC2) significantly reduce the correspondence (Fig. 2a, left), suggesting that PC2 alone encoded the size feature (Fig. 2a, right. PC2: DI = 1.34, $p < 0.05$, Bonferroni corrected; the rest: DIs < 0.01, ps > 0.05; Supplementary Data 2, 3). This finding was observed in other layers of AlexNet (Fig S4), and other DCNNs tested (Fig S5a). In short, the feature of objects' real-world size was specifically encoded by one axis of the object space alone.

A great challenge to encode objects' size is that the size varies greatly (e.g., airplanes are 2–3 orders of magnitude larger than basketball) and the size of daily objects is in a heavy tail distribution, with the concentration mainly in the range of centimeters to meters. To examine how this axis efficiently encoded objects' real-world size, we tested a variety of encoding schemes, such as linear, power, and logarithm functions. Among all functions examined, the best function that maps the physical world (i.e., real-world size) to the representational space (i.e., values of PC2) was the common logarithm scale ($R^2 = 0.48$, $p < 0.001$, Fig. 2b; Supplementary Data 2). That is, the stimulus-representation mapping follows the Weber-Fechner law, suggesting that AlexNet compresses large physical intensity ranges (i.e., objects' real-world size) into smaller response ranges of units. Similar results were also found in other DCNNs (Fig S5b).

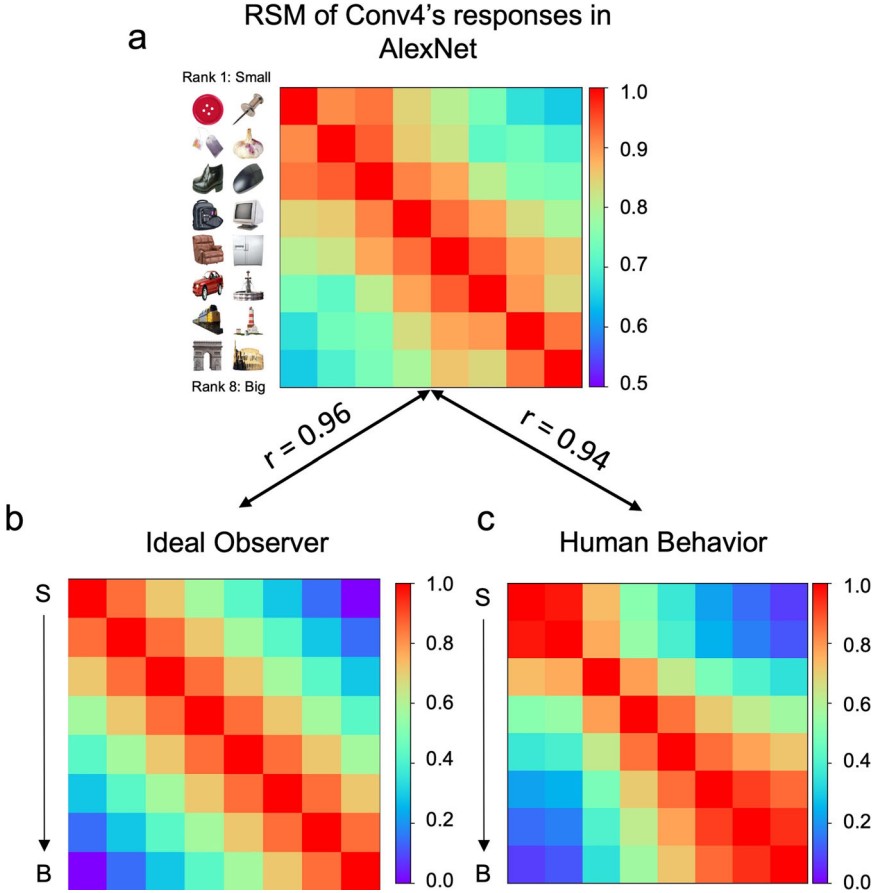

**Fig. 1 Representation of objects' real-world size emerged in AlexNet designated for object recognition. a** The RSM of units' responses at Conv4 layer to objects with different sizes, and objects with similar sizes elicited similar response patterns indexed by warm colors. **b** The RSM of the ideal observer and **c** human showed high correspondence to that of AlexNet. S small objects, B big objects.

Behavioral studies on humans have revealed that the sensitivity to objects' size significantly facilitates object recognition[13]. Accordingly, here we explicitly examined the role of the size axis in object recognition with an ablation analysis that is not applicable in biological systems. That is, we removed Conv4's response variance aligned to this axis to examine whether AlexNet's performance on object recognition was impaired. We found that with the residue responses after removing PC2, AlexNet's Top-1 accuracy of recognizing the ImageNet validation images was slightly but significantly decreased from 52.6% to 48.5% ($p < 0.001$) (Fig. 2c, left), indicating that the size axis contributed to object recognition. This finding is consistent with the facilitation of objects' size in recognizing objects observed in humans. Similar results were found in other DCNNs (Fig S5c) as well.

**Objects' shape and texture were used to infer objects' size in DCNNs.** As expected, the size axis was absent in an untrained AlexNet (Fig S3), showing the necessity of stimulus experience, rather than DCNNs' architecture, in constructing the size axis. To further explore factors that may contribute to the construction of the size axis, we first examined the factor of object co-occurrence (i.e., context) that provides the relative difference in retinal size among objects. That is, in natural environment, objects are seldomly present alone, and the co-occurrence of multiple objects in an image likely provides information on relative size among objects (e.g., the person in Fig. 3a is smaller than the car), which in turn may be used to infer objects' real-world size. To examine this possibility, we trained a new AlexNet with images containing only one single object without any background (i.e., the single-

object AlexNet). Surprisingly, the single-object AlexNet was still able to represent objects' real-world size, supported by a close correspondence of Conv4's responses to the ideal observer (Fig. 3b; $r = 0.96$, $p < 0.001$; Supplementary Data 1) and an axis specifically encoding the size feature in object space (Fig. 3c, DI = 0.87, $p < 0.05$, corrected; Supplementary Data 3) with a common logarithm function ($R^2 = 0.37$, $p < 0.001$; Supplementary Data 2). In addition, we directly manipulated the absolute retinal size of the objects from the real-world size dataset, and the representation of objects' real-world size remained unchanged (Fig S6). Therefore, neither the relative nor absolute retinal size of objects contributed to the construction of the size axis, ruling out the possibility of context in extracting the feature of objects' size.

On the other end of DCNNs are task demands, which have been demonstrated to modulate the representation of DCNNs[23,25]. To test top-down task demands on the representation of objects' size, we trained two new AlexNets with the same image datasets but to differentiate objects at a coarse level of living things versus artifacts (i.e., the AlexNet-Cate2) or at a superordinate level of 19 categories (i.e., the AlexNet-Cate19, see Methods for all superordinate categories; Fig. 4a). Again, the task demands had little effect on the representation of objects' size, with the best correspondence to the ideal observer in Conv4 (Fig. 4b; AlexNet-Cate2: $r = 0.95$, $p < 0.001$; AlexNet-Cate19: $r = 0.96$, $p < 0.001$; Supplementary Data 1) and the same axis encoding objects' size (AlexNet-Cate2: DI = 1.30, $p < 0.05$, corrected; AlexNet-Cate19: DI=1.30, $p < 0.05$, corrected; Supplementary Data 3) with a common logarithm function (AlexNet-Cate2: $R^2 = 0.37$, $p < 0.001$; AlexNet-Cate19: $R^2 = 0.46$, $p < 0.001$) (Fig. 4c; Supplementary Data 2).

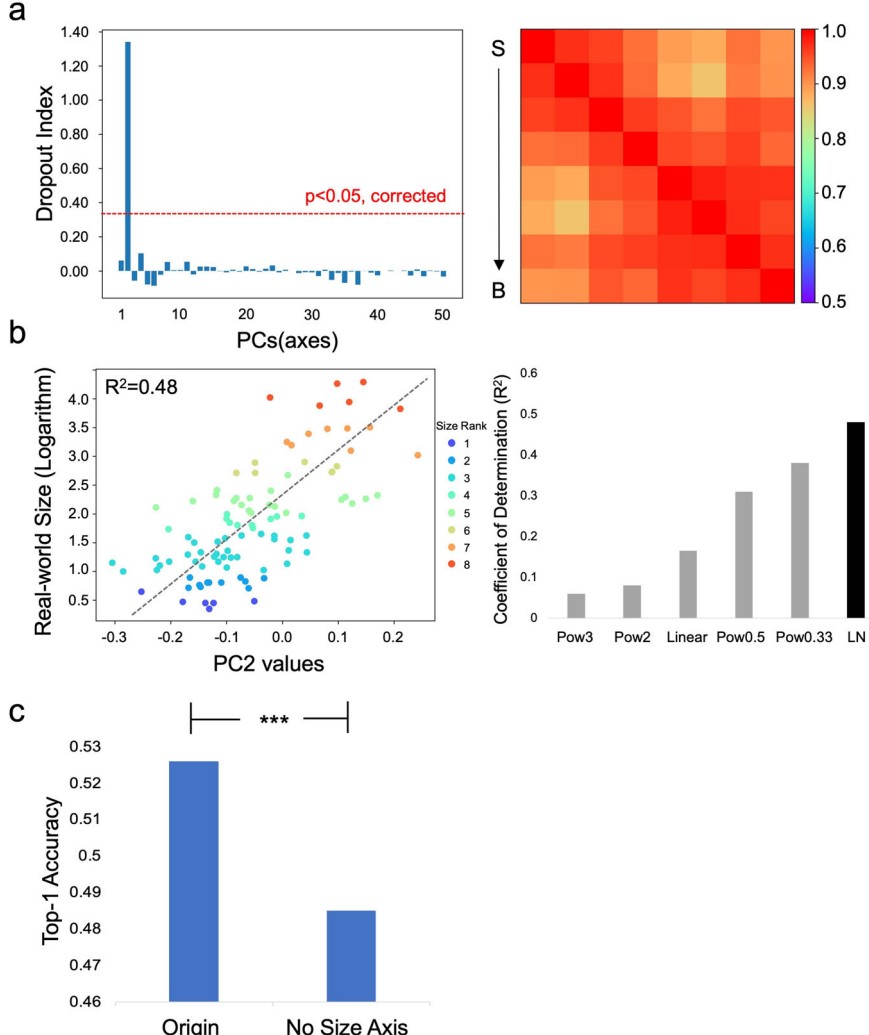

**Fig. 2 The axis of object space for objects' real-world size. a** Among all 50 PCs, only the removal of PC2 significantly reduced the correspondence between the RSM of Conv4's responses and that of the ideal observer (Left). This is demonstrated by the RSM of Conv4's residual responses after regressing out response variance aligned to PC2, which no longer showed sensitivity to objects' size (Right). **b** The mapping of objects' real-world size to PC2 was best described as a common logarithm function (left), among all functions tested (right). The color indicates size rank. **c** The removal of the response variance aligned to PC2 significantly impaired AlexNet's performance on object recognition. S small objects, B big objects. Pow power, LN common logarithm.

Since factors other than objects' own properties apparently had little contribution to DCNNs in extracting the feature of objects' size to construct object space, here we tested whether objects' own properties, such as shape and texture suggested by previous studies on humans[13,28], were critical. Three types of stimuli were created from the real-world size dataset: silhouette, texture and shuffle images (Fig. 5a). To thoroughly decouple the effect of shape and texture in representing objects' size, silhouette images were adopted, which preserve overall shape information with no texture information, whereas texture images, on the other hand, preserve only texture information. Shuffle images were used to serve as a control condition, which contains neither shape nor texture information. We found that AlexNet was able to infer objects' size with either shape ($r = 0.89$, $p < 0.001$) or texture information ($r = 0.73$, $p < 0.001$) presented alone (Fig. 5b; Supplementary Data 1), and the size feature was also encoded in PC2 of object space (silhouettes: $R^2 = 0.34$, $p < 0.001$; textures: $R^2 = 0.33$, $p < 0.001$; Fig. 5c; Supplementary Data 2). In contrast, when both the shape and texture information were scrambled (i.e., shuffle images), the AlexNet was no longer able to extract the size feature ($R^2 = 0.01$, $p > 0.05$). Taken together, DCNNs seem

to use objects' shape and texture, rather than external factors of objects' absolute retinal size, context, and task demands, to infer objects' size.

**The objects' shape, not texture, was necessary for both DCNNs and humans to infer objects' size**. The finding that either shape or texture of objects alone was sufficient to infer objects' size in DCNNs echoes neuroimaging studies in human that texform stimuli, which preserve both texture and shape information but are not recognizable, can successfully recapitulate objects' size[14,29]. However, behavioral studies on humans show that texture alone is not able to infer objects' size[13], which is apparently contrary to the role of texture observed in DCNNs. To further explore the role of texture on objects' size in humans, we asked whether the texture shown to AlexNet could activate cortical regions with size sensitivity in the VTC. Specifically, we used fMRI to scan human subjects when they passively viewed the objects from the real-world size dataset and their silhouettes and textures. First, we replicated the medial-to-lateral arrangement of a big-to-small map separated by the MFS when the original

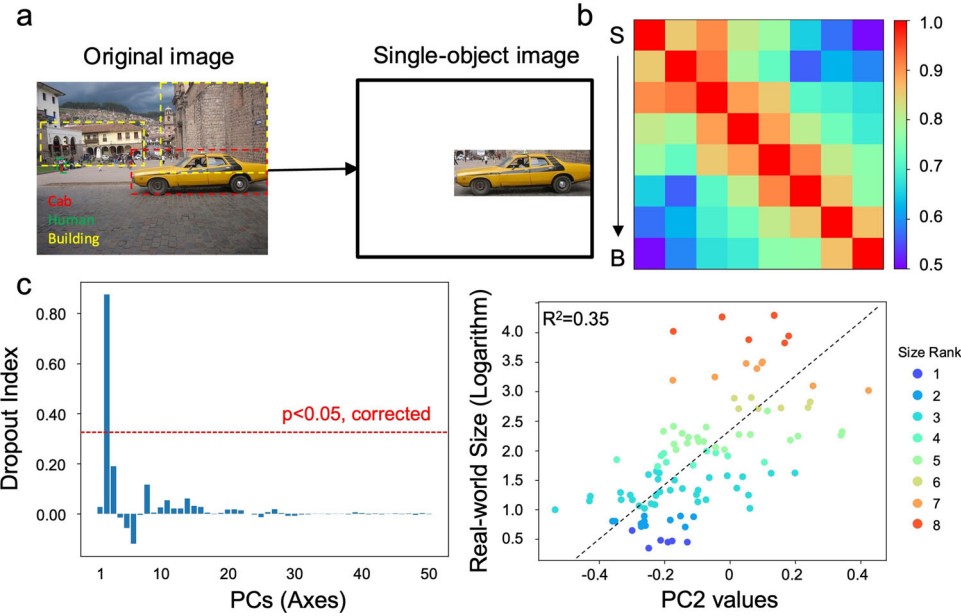

**Fig. 3 Factors of objects' retinal size in inferring objects' real-world size. a** Relative differences in retinal size among objects. In natural images, objects with smaller retinal size (e.g., persons) are usually smaller than those with larger retinal size (e.g., cars, buildings). The effect of this factor was tested by removing the background. **b** The RSM of Conv4's responses in the single-object AlexNet to objects' real-world size. **c** The second axis in object space specifically encoded the real-world size, and the best mapping function was the common logarithm. S small objects, B big objects.

objects and the silhouettes (i.e., shape) were presented (Fig. 6a, top and middle panels). However, the textures failed to activate the VTC (Fig. 6a, bottom panel). This observation was further quantified by a region-of-interest (ROI) analysis, where the size-sensitive ROI was pre-defined by the contrast of big versus small objects. We found in the ROI a significantly higher BOLD response to big objects in the silhouette condition ($t = 4.92$, $p < 0.001$), but not in the texture condition ($t = 1.14$, $p = 0.27$) (Fig. 6b), suggesting the size-sensitive cortical region in human used the shape information, but not texture information, to infer objects' size.

This finding was further confirmed by a finer multivariate pattern analysis, where the activation pattern in the ROI was only able to distinguish big objects from small objects in the silhouette condition (classification accuracy = 0.80, $p < 0.05$; Fig. 6c) but not in the texture condition. A thorough whole-brain searchlight analysis did not find any region capable of distinguishing objects' size based on texture information (Fig. 6d). In short, the finding from the fMRI experiment is in line with previous studies that when shape information is removed from texform[13] or when images are globally scrambled[28,30], the neural activation in the VTC is no longer similar to that of their original ones. That is, the human brain apparently relied extensively more on shape than texture to infer object's size.

The difference between humans and DCNNs in relying on texture information to infer objects' size may reflect the fact that DCNNs are heavily biased by objects' texture[31,32], which primarily originates from the training data[33]. Therefore, it is not surprising that DCNNs used texture information to infer objects' size as well. However, the finding from human's VTC raised an interesting question of whether texture information is necessary for DCNNs to infer objects' size, given that texture information is unstable in the natural environment, and largely affected by climates, air, or illumination. To address this question, here we used the stylized AlexNet[32], which is trained on a stylized version of the ImageNet (i.e., Stylized-ImageNet). With randomly selected painting styles (Fig. 7a), objects' shape and texture are decoupled as objects' texture is significantly distorted. We found

that the stylized AlexNet showed a similar pattern as the original AlexNet, where Conv4's responses showed sensitivity to objects' real-world size (correspondence with the ideal observer: $r = 0.86$, $p < 0.001$; Fig. 7b; Supplementary Data 1) and only one axis of object space encoded the size feature (Fig. 7c; DI = 0.45, $p < 0.05$, corrected; Supplementary Data 3) with the mapping function of common logarithm ($R^2 = 0.29$, $p < 0.001$) (Fig. 7d; Supplementary Data 2). Importantly, this size axis was much less sensitive to the texture information ($R^2 = 0.08$, $p < 0.001$ for texture images) as compared to the shape information ($R^2 = 0.22$, $p < 0.001$ for silhouette images). Taken together, the visual experience of correct texture information of objects was not necessary to infer objects' size, implying a strong equivalence between DCNNs and humans in representing objects' real-world size.

**Object's size was independent encoded from curvature and animacy in object space.** Both the fMRI and DCNN experiments suggest the critical role of shape in extracting the feature of objects' size. Among all types of shapes, curvature (spiky versus stubby) is most related because it is an axis of object space[9] and a mid-level stimulus property that may provide important information for objects' size (e.g., big objects are boxier but small objects curvier). Therefore, it is possible that curvature and objects' size may share the same axis of object space. To test this possibility, we measured objects' curvature by calculating objects' aspect ratio[9], with larger values indicating spiky objects and small values for stubby objects. We found that there was no significant correlation between curvature and real-world size of objects (Fig. 8a; $R^2 = 0.01$, $p = 0.41$). Besides, the loading of curvature on PC2 was small (Fig. 8b; $R^2 = 0.001$, $p = 0.72$), which was much smaller than the loading of objects' size ($R^2 = 0.48$), suggesting that the size axis unlikely relied on curvature as an important shape property to represent objects' size. Another well-established axis of object space is animacy (animate versus inanimate), which forms a tripartite organizational schema with objects' size (i.e., big artifacts, animals, and small artifacts)[7]. Consistent with the observation in human, the size axis was not sensitive to the size of animals (Fig. 8c; $t = 1.78$, $p = 0.08$). Instead, the feature of animacy was apparently encoded in the first

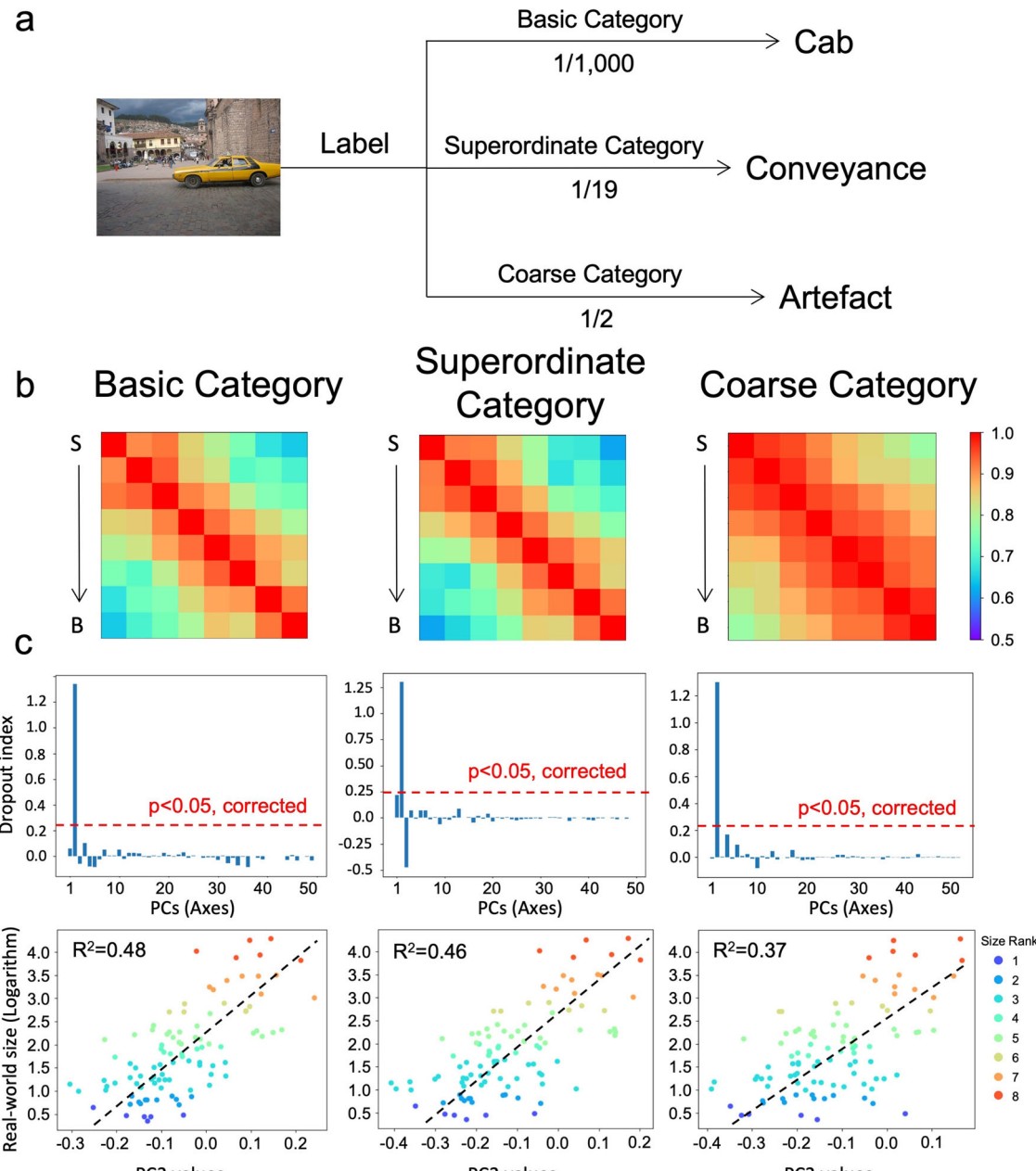

**Fig. 4 Factors of objects' task demands in inferring objects' real-world size. a** Task demands. Three DCNNs with the same architecture (i.e., AlexNet) were trained to classify objects at different levels of categorization. DCNNs were trained to categorize a car as cab (basic level), conveyance (superordinate level), or artifact (coarse level). Note that the DCNN for the basic-level categorization is the same as the one used in the previous experiments. **b** The RSM of Conv4's responses of the AlexNets with different task demands. From left to right: basic level, superordinate level, and coarse level. **c** The second axis of the object space also specifically encoded the real-world size with the mapping function of the common logarithm. S small objects, B big objects.

principal component (PC1) of the object space, as the values of PC1 distinguished animacy from artifacts (Fig. 8d; $t = 13.05$, $p < 0.001$). In sum, the feature of objects' size was independent of the feature of curvature and animacy in the object space.

## Discussion

In this study, we used both DCNNs and fMRI to examine how objects' real-world size serves as an axis of object space. We showed that DCNNs designated for object recognition automatically extracted the feature of objects' real-world size and used it for object recognition. This size feature was mapped to one axis of object space through nonlinear transformation of the common logarithm. Because the DCNNs relied purely on the experience of

perceiving objects, the emergence of the size axis in the DCNNs suggests that experience-dependent mechanisms of detecting visual image statistics are sufficient for constructing the size axis without innately predisposed sensitivity to objects' size. Further in silico experiments revealed that the intrinsic properties of the objects (i.e., shape and texture), rather than the external factors such as absolute retinal size, context, or task demands, were the key factors for DCNNs to infer objects' size. Finally, echoing the fMRI finding that humans relied extensively more on shape than feature to infer objects' size, we also found that in DCNNs the visual experience of correct texture information of objects was not necessary to infer object's size. Taken together, with DCNNs as a brain-like model, we showed how objects' real-world size serves

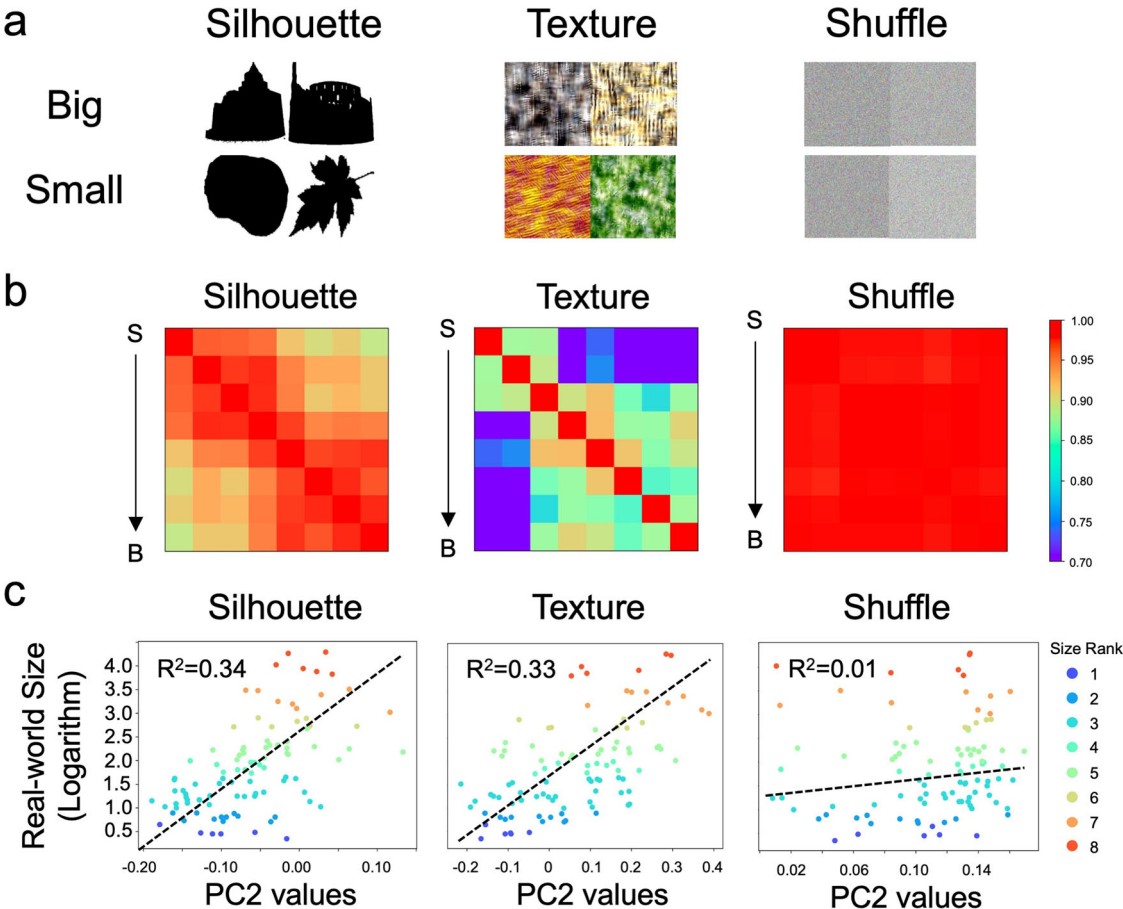

**Fig. 5 The factors of objects' shape and texture in inferring objects' size. a** Image examples of silhouette, texture, and shuffle conditions. **b** The RSM of Conv4's responses to objects' size. The sensitivity to the size was found when either silhouettes or textures were presented, but not when shuffle images were presented. **c** The common logarithm mapping between PC2 and objects' size was found in both silhouettes and textures, but not in shuffle conditions. S small objects, B big objects.

as an axis of object space, which provides computational support for empirical observation on this feature in human perceptual and neural representations of objects.

Object space has been proposed as a computational framework to efficiently encode objects by critical object features that construct axes of object space[1,2,34–36]. Under the view of vision for perception, previous studies mainly focus on the physical attributes of objects, such as physical appearance (e.g., having furs, wings), curvature (e.g., spiky, stubby), and conceptual knowledge (e.g., animate, artifacts) as candidate features for axes. In this study, we focused on the feature of objects' real-world size that not only facilitates object recognition (vision for perception) but also heuristically affects our action on objects (vision for action)[37]. We found that DCNNs solely designated for perception automatically encoded the feature of object's size, immune to context and action-based task demands, implying that the perceptual analysis of objects' shape was likely the main source for our brain to develop the sensitivity to objects' size, to infer objects' relation to the environment, and to find appropriate actions upon objects. In fact, neuropsychological studies on patients reveal that objects' shape was used for actions on the objects, though the patients were not consciously aware of the objects[38–40], and our fMRI experiment, along with previous studies[13,14,29], also showed that shape information alone (i.e., silhouettes) was sufficient to activate the size-sensitive regions in the VTC. Therefore, the perceptual analysis of objects' shape seems a pre-requisite for action on the objects.

In addition, our study also revealed two characteristics of the size axis, which may provide insights into other axes of object space. First, we found that the size axis, like axes for animacy and curvature[7,9,11,12], was statistically orthogonal to other axes. This is important, because from the perspective of the resource rationality[41,42], it is uneconomical if object features used for constructing object space are redundant and correlated. Therefore, orthogonality among axes of object space is likely to reduce the redundancy of visual details effectively[43,44]. Second, the mapping from objects' real-world size to the size represented in the size axis was not linear; instead, the coding scheme adopted by the size axis followed the Weber-Fechner Law that the scale for large size was compressed exponentially, similar to the nonlinear coding strategy adopted by human[45]. The advantage of the compressed scaling is likely to enlarge coding space to avoid explosions in the number of preferred neurons needed[46] and to increase the tuning range of neurons[47] so that they can efficiently encode objects with sizes differing in order of magnitude. With these two characteristics, objects' real-world size is encoded efficiently in object space.

In sum, with brain-like DCNNs, our study revealed how objects' real-world size was extracted and used to construct an axis of object space. However, several unresolved issues need future studies. First, in this study we only focused on objects' size, and it is interesting to know how this feature works collaboratively with other features to construct object space to represent objects as a whole. Second, objects' shape was found as a key

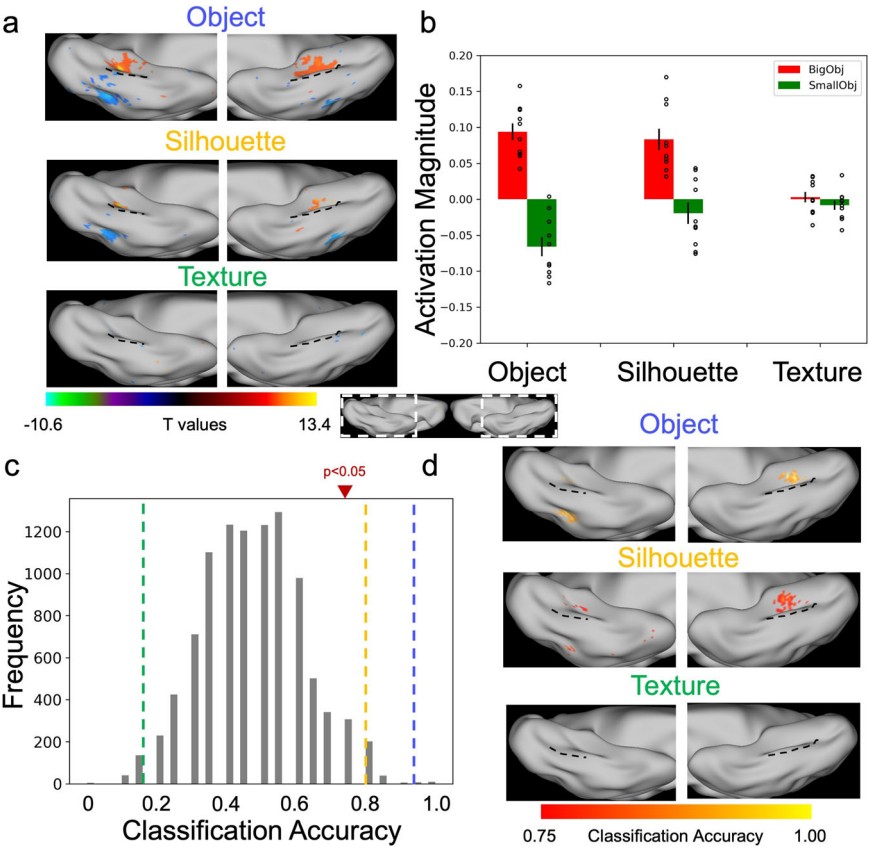

**Fig. 6 Objects' shape and texture in inferring objects' size in human brain. a** Size-preference activation maps (big vs. small) from the group averaged fMRI data when objects and their silhouette and texture were present. A medial-to-lateral arrangement of the size preference in the VTC was present in the object and silhouette conditions, but not in the texture condition. Black dotted line: MFS. **b** Activation magnitude showed a significant difference between big and small objects in the size-sensitive ROI in the object and silhouette conditions, but not in the texture condition. **c** The activation patterns in the ROI to objects and silhouettes, but not texture, were capable of distinguishing big from small objects. Blue line: objects; Yellow line: silhouettes; Green line: texture. **d** A whole-brain searchlight analysis found regions capable of distinguishing big from small objects only in the VTC and only for objects and silhouettes, but not for texture.

factor to infer objects' size; however, in the study, we only tested one mid-level perceptual property of curvature, and found that it had little contribution in representing objects' size. Future studies need to explore which mid-level properties, such as local corners, junctions, and contours, embedded within shape information, provide real-world size distinctions between objects. Finally, because of the discriminative nature of DCNNs, in this study we only explored the role of bottom-up factors, such as shape and object co-occurrence, to infer objects' sizes, which are unlikely to differentiate the size of toy cars from that of real cars. Future studies with generative models on top-down factors, such as semantic interference and action-based task demands, may help understand how objects' size is represented comprehensively.

## Materials and Methods
**Neural network models.** Multiple pre-trained DCNNs were used in this study.

The AlexNet[27] includes 8 layers of computational units stacked into a hierarchical architecture; the first 5 layers are convolutional layers and the last 3 layers are fully connected layers. The second and fifth convolutional layers are followed by the overlapping max-pooling layers, while the third and fourth convolutional layers are connected directly to the next layer. Rectified linear unit (ReLU) nonlinearity was applied after all convolutional and fully connected layers. Layer 1 through 5 consisted of 64, 192, 384, 256, and 256 kernels.

Several extra DCNNs, including VGG11, VGG13, ResNet18, ResNet34, and Inception_v3, were used to verify whether results from the AlexNet could be replicated in other network architectures.

Two VGG networks[48], VGG11 and VGG13, were used to examine the effect of layer numbers on the formation of real-world size preference in DCNNs. The VGG11 and VGG13 include 11 and 13 layers respectively, with the first 8 and 10 layers being convolutional layers and the last 3 layers being fully connected layers. All hidden layers are equipped with the ReLU nonlinearity. For VGG11, overlapping max-pooling layers follow the 1, 2, 4, 6, 8 convolutional layers. For VGG13, overlapping max-pooling layers follow the 2, 4, 6, 8, 10 convolutional layers.

Two ResNet networks[49], ResNet18 and ResNet34, were used to examine the effect of residue blocks on the formation of real-world size preference in DCNNs. ResNet18 and ResNet34 include 18 and 34 layers respectively, with all layers being convolutional layers except for the last one being a fully connected layer. A residue block was constructed between every two convolutional layers by inserting a shortcut connection. All hidden layers are also equipped with the ReLU nonlinearity.

Inception_v3[50] was used to examine the effect of inception structure on the formation of real-world size preference in DCNNs. Inception_v3 includes 5 independent convolutional layers, 10 Inception modules, and 1 fully connected layer in total. Each Inception module consists of several convolutional layers with small kernel sizes arranged in parallel. The 10 Inception modules could be classified into InceptionA, InceptionB, InceptionC, InceptionD or InceptionE modules. Among the 10 Inception modules, three are InceptionA modules, followed by one InceptionB module, four InceptionC modules, one InceptionD module and two InceptionE modules. The detailed architecture could be referred to[50].

Each neural network model was pre-trained to perform object classification on the ILSVRC2012 ImageNet dataset[51], which includes about 1.2 million images of objects belonging to 1,000 categories. The object classification accuracy was evaluated on 50,000 validation images that were not seen by the model during training. The Top-1 and Top-5 accuracies of AlexNet are 52.6% and 75.1%. The network weights of all neural network models were downloaded from the PyTorch model Zoo (https://pytorch.org/vision/0.8/models.html)[52].

A stylized AlexNet[32], which has the same architecture as the classical AlexNet but is trained with a Stylized-ImageNet dataset (SIN) was downloaded from https://github.com/rgeirhos/texture-vs-shape/tree/master/models. The SIN was constructed by replacing the style (i.e., texture) of images from the ImageNet ILSVRC2012 training dataset with styles of different paintings (see Fig. 5a for

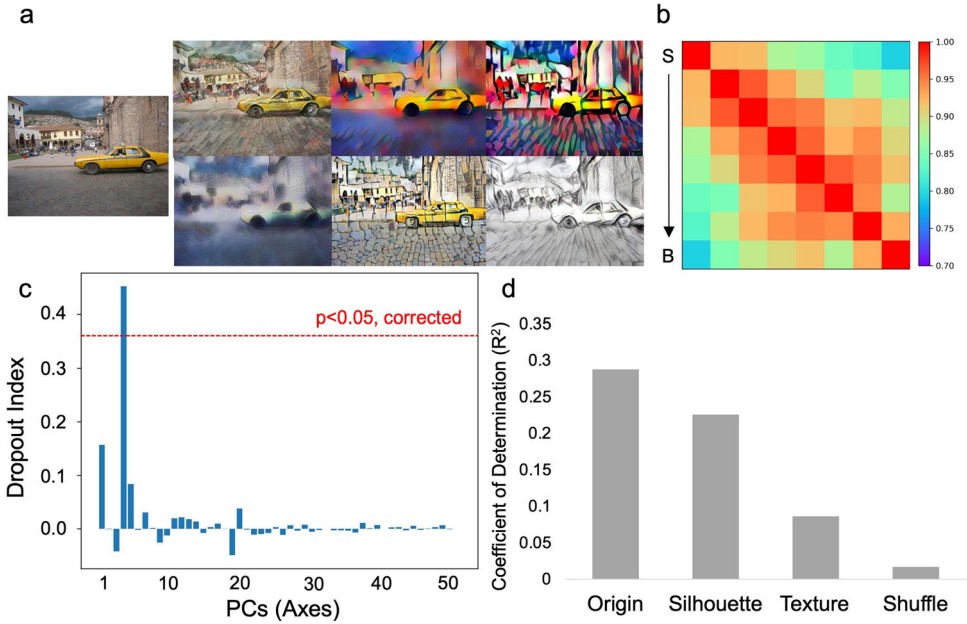

**Fig. 7 The necessity of texture information in inferring objects' size in DCNNs. a** Images from the ImageNet were transformed with various painting styles, where the shape information was preserved and the texture information distorted. **b** The RSM of Conv4's response to objects' size is similar to the ideal observer. **c** Only the fourth axis of object space constructed by the stylized AlexNet encoded the size feature. **d** Logarithm correspondence between PC4 and objects' size showed much less sensitivity to texture as compared to shape.

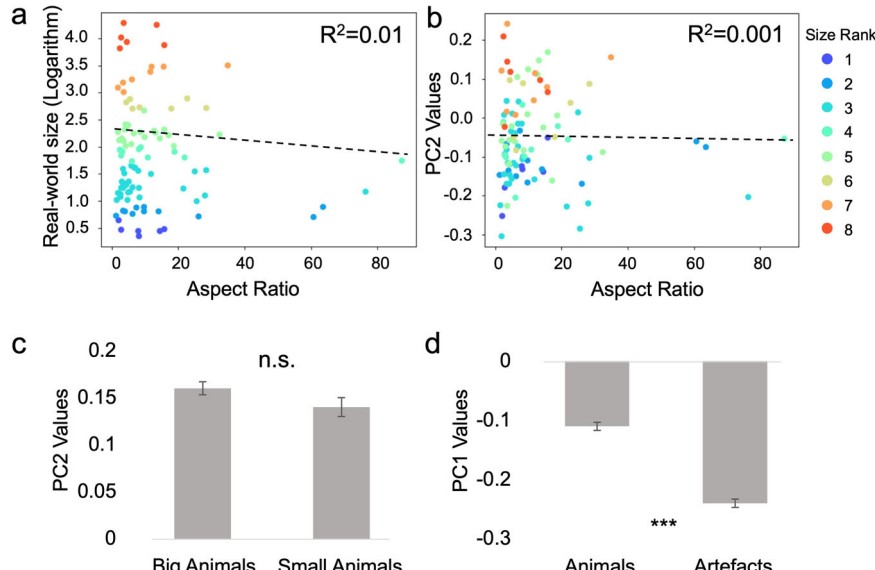

**Fig. 8 The independence of coding objects' size, curvature and animacy in the object space.** Curvature was measured by aspect ratio, which showed no significant correlation with either objects' size (**a**) or PC2 (**b**). **c** The size axis (PC2) was not sensitive to the size of animals ($t = 1.78$, $p = 0.08$). **d** Instead, the feature of animacy was apparently encoded in the first principal component (PC1) of the object space, as the values of PC1 distinguished animacy from artifacts ($t = 13.05$, $p < 0.001$). ***$p < 0.001$; n.s. not significant.

examples). The object recognition ability of the stylized AlexNet was mainly achieved via shape but insensitive to texture variance of the object images.

**Stimulus datasets and model training**. To examine whether real-world size preference emerges in the AlexNet pre-trained for object classification, we used a real-world size dataset downloaded from https://konklab.fas.harvard.edu/ImageSets/OBJECT100Database.zip, which was previously used[45] to evaluate real-world size preference in humans. This dataset includes 100 background-free object images spanning the range of real-world size from small objects (e.g., thumbtack) to large objects (e.g., colosseum). Each image consists of a single object, labeled with the measured real-world size of this object. The real-world size of each object was measured as the diagonal of its bounding box, ignoring the depth of the object, and quantified in centimeters.

Silhouette, texture, and shuffle versions of the real-world size dataset were separately generated to examine the contribution of shape and texture information to the representation of real-world size in AlexNet. In detail, to generate a silhouette of the original object, we first detect the edges of an object using the canny edge detection algorithm, then set all values within edges as 0 (i.e., black color); this removed texture information from the original objects. Curvature of objects was measured using the aspect ratio, which is defined as a function of perimeter $P$ and area $A$:

$$\text{Aspect Ratio} = \frac{P^2}{4\pi A} \qquad (1)$$

where $P$ was measured by the number of pixels lying on the object's contour, $A$ was measured by the number of pixels of the silhouette.

On the other hand, a texture version of the original object was synthesized using the Portilla–Simoncelli algorithm (https://github.com/LabForComputationalVision/textureSynth)[53]. In detail, the Portilla–Simoncelli algorithm obtained four sets of parameters from the target image and altered a random-noise image into a synthesized texture image by iteratively matching its parameter distributions with those of the target image. The four sets of parameters included (1) a series of first-order constraints on the pixel intensity distribution, (2) the local autocorrelation of the target image's low-pass counterparts, (3) the measured correlation between neighboring filter magnitudes, and (4) cross-scale phase statistics. This procedure ensured that the random-noised images converged as a texture counterpart of the original images but reserved no shape information. The shuffle format of the real-world size dataset was generated by randomly shuffling the original images, which destroyed all shape and texture information of the object.

To evaluate whether the retinal size difference among objects accounted for real-world size preference of the AlexNet, we re-trained the AlexNet from scratch with a single-object version of the original ImageNet dataset that contains no background information. To do this, we first downloaded annotations of object bounding boxes from http://image-net.org/download-bboxes, which were annotated and verified through Amazon Mechanical Turk. 544,546 bounding boxes are corresponding to the original training images and 50,000 bounding boxes correspond to the original validation images, respectively. We removed the background of each image by setting pixels outside the bounding box to 255 (i.e., white color). For images containing multiple bounding boxes, we randomly selected one bounding box as our target. Note that the retinal size of objects remained unchanged with only the background removed from the original images. The AlexNet was trained for 50 epochs, with an initial learning rate of 0.01 and a step multiple of 0.1 in every 15 epochs. Parameters of the model were optimized using stochastic gradient descent with the momentum and weight decay fixed at 0.9 and 0.0005. The Top-1 and Top-5 accuracies of the AlexNet that trained with the single-object image dataset (i.e., the single-object AlexNet) were 46.7% and 72.0%, respectively.

To evaluate the effect of task demand on the emergence of the real-world size axis in object space from the AlexNet, we separately re-trained two AlexNets from scratch. One was to classify objects into two coarse categories, the living things and artifacts (the AlexNet-Cate2), the other was to classify objects into 19 superordinate categories (the AlexNet-Cate19), including fungus, fish, bird, amphibian, reptile, canine, primate, feline, ungulate, invertebrate, conveyance, device, container, equipment, implement, furnishing, covering, commodity, and structure suggested by WordNet[54]. All object images were selected from the ImageNet dataset, which consisted of 866 categories in total. The number of images for training is 1,108,643, and for validating is 43,301. The AlexNet-Cate2 and AlexNet-Cate19 shared the same architecture as the original AlexNet, except that we added one extra FC layer after the FC3 layer for the classification of two coarse or 19 superordinate categories. The AlexNet-Cate2 and AlexNet-Cate19 were trained following the same procedure as the single-object AlexNet. The Top-1 and Top-5 accuracies of the AlexNet-Cate2 were 94.7% and 100.0%, and the Top-1 and Top-5 accuracies of the AlexNet-Cate19 were 68.7% and 95.6%, respectively.

To investigate the relationship between the size axis and objects' animacy, an animacy-size dataset was downloaded from https://konklab.fas.harvard.edu/ImageSets/AnimacySize.zip, which contains background-free objects of big animals, big artifacts, small animals, and small artifacts. For each type of object, the dataset included 60 images, which consists of 240 images in total.

## Calculate representational similarity matrix (RSM) of real-world size in AlexNet.
Representational similarity between objects in different real-world sizes was used to evaluate whether real-world size preference automatically emerges in a DCNN that is trained for object recognition.

To achieve that, we extracted responses to objects from the real-world size dataset in different layers of the AlexNet. All images were transformed with resize and normalization to match the input requirement of the AlexNet. No ReLU was performed for the responses. We averaged responses from the convolutional layers within each channel, resulting in a response pattern of 256 channels in Conv4 for each image. We grouped objects into eight size ranks according to their real-world sizes (see Table S1). Each size rank included no less than six objects in different viewpoints, colors, and object shapes, to balance unrelated confounding factors. We averaged object response patterns within each size rank, and calculated similarity between these averaged response patterns in different size ranks to examine whether the response patterns in nearer size ranks are more similar with each other than those further apart, resulting in an RSM of size ranks in the AlexNet.

The degree of real-world size preference was quantified by comparing it with an ideal observer model. The RSM of the ideal observer model was constructed by the consistency between each pair of size ranks, which was defined as follows:

$$\text{Consistency}_{ij} = 1 - \left| \frac{\text{Rank}_i - \text{Rank}_j}{8} \right| \tag{2}$$

where $i$ and $j$ are indicators to denote different size ranks, $\text{Rank}_i$ and $\text{Rank}_j$ are size ranks, which ranged from 1 to 8. High correspondence between the RSMs of the AlexNet and the ideal observer suggested real-world size preference emerged in the AlexNet.

We also measured the real-world size representation of humans. Two participants (two males; 22 and 27 years) were recruited to separately compare the sizes of the objects from the real-world size dataset. For each pair of objects, participants were required to indicate which object is bigger than the other. Each participant completed 4,950 comparisons (i.e., $C_{100}^2$) for all pairs of the 100 images, which provided a proportional value for each object with the following formula:

$$\text{Prop}_i = \frac{\sum_{j=1}^{100} \mathbb{I}_{ij}}{100} \tag{3}$$

where $\mathbb{I}_{ij}$ equals to 1 when the $i$th object is judged to be bigger than the $j$th object. An object with a higher proportional value indicated a larger real-world size in a human's mind. The proportional value of a size rank was calculated as the averaged proportional value across objects belonging to the same size rank. The RSM of humans was constructed by the consistency between each pair of size ranks measured as

$$\text{Consistency}_{ij} = 1 - \left| \frac{\text{Prop}_i - \text{Prop}_j}{8} \right| \tag{4}$$

where $\text{Prop}_i$ and $\text{Prop}_j$ are proportional values of the $i$th and $j$th size rank.

**Evaluate the role of real-world size in object space**. To evaluate whether real-world size played a role as a principal axis in object space, we used Principal Component Analysis (PCA) to recover an object space using images from the ImageNet validation dataset. Specifically, we first fed all 50,000 validation images into a pre-trained AlexNet. Responses from the layer with the highest real-world size preference (i.e., Conv4) were extracted and then averaged within each channel to generate a response matrix (Image numbers × Channel numbers). We further normalized it by dividing its second-order norm across images. Then we used PCA to decompose the response matrix of the AlexNet into multiple principal axes. This yielded a linear transformation between responses and principal components (PCs) as follows:

$$C = X \times V \tag{5}$$

where $C$ is the PCs, $X$ is the responses and $V$ is the principal axes in object space. We retained the first 50 principal axes, which captured 94.7% of the variance in the response of the AlexNet (More than 90% for other DCNNs).

We further investigated whether the real-world size was represented as a principal axis in object space. We first extracted responses to images from the real-world size dataset in the same layer that was used for the construction of principal axes, and got the projected principal components. To evaluate the effect of each axis on the real-world size preference, we iteratively removed the variance of each component from the original responses, and re-calculated the RSM of real-world size in the AlexNet. Reduction of correspondence to the ideal observer was measured with a dropout index (DI) defined as follows:

$$DI_i = Z(R) - Z(r_i) \tag{6}$$

where $r_i$ is the correspondence between the real-world size RSMs of AlexNet and the ideal observer after removing the $i$th PC, and R is the correspondence without removing any PCs. Z(*) is the Fisher z-transformation. The higher value indicated the larger effect of a PC on the real-world size preference.

The significance of DIs was evaluated by comparing it with a null distribution, which was generated from an untrained AlexNet with repetition of the same procedures 5000 times for each principal axis. Multiple comparison correction was performed using Bonferroni correction after considering an integrated null distribution across all principal axes. The PC with a significant DI value was identified as the axis encoding real-word size.

The quantitative relationship of this axis to the real-world size of objects was evaluated with a series of functions in different scales:

$$\text{Linear}: y = x \tag{7}$$

$$\text{Exponential}: y = x^{0.33} \tag{8}$$

$$\text{Exponential}: y = x^{0.5} \tag{9}$$

$$\text{Exponential}: y = x^2 \tag{10}$$

$$\text{Exponential}: y = x^3 \tag{11}$$

$$\text{Logarithmic}: y = \log_{10} x \tag{12}$$

where $x$ is the value of the real-world size PC, and $y$ is the measured real-world size of objects. The scale with the best fit was taken as the relationship that best describes the data.

The role of the size axis in object recognition was evaluated with an ablation analysis by examining the impairment of the Top-1 accuracy from the last fully connected layer. Similar to the measurement of DI, we first removed the variance of each PC from the original responses in Conv4, and then fed these responses into higher layers to get the Top-1 accuracy of the AlexNet for each category. Significance was tested by comparing it with a null distribution, which was

generated from baseline object spaces originating from an untrained AlexNet with a repetition of 1000 times.

## fMRI Experiments

*Participants*. 10 participants (5 males, age range: 18–27 years) from Beijing Normal University participated in this study to examine the effects of shape or texture on the real-world size representation in the human brain. All participants had a normal or corrected-to-normal vision. Informed consent was obtained according to procedures approved by the Institutional Review Board of Beijing Normal University.

*Image Acquisition*. Imaging data were collected on a 3T Prisma Siemens MRI Scanner with a 64-channel phased-array head coil at Beijing Normal University Imaging Center for Brain Research. The anatomical images were acquired with a magnetization-prepared rapid gradient-echo (MPRAGE) sequence. Parameters for the T1 image are: TR/TE = 2530/2.27ms, flip angle = 7°, voxel Resolution = 1 × 1 × 1mm. Blood oxygenation level-dependent (BOLD) contrast was obtained with a gradient-echo-planar T2* sequence. Parameters for the T2* image are: TR/TE = 2000/34.0ms, flip angle = 90°, voxel Resolution = 2 × 2 × 2 mm, FoV = 200 × 200 mm.

*Experiment design*. All participants completed eight runs of fMRI scanning. Participants were shown images of big or small objects in a standard block design. All objects were displayed at the same visual angle (5.3° × 5.3°, visual distance = 100cm) to exclude the confounding effect of different retinal sizes of objects. Big objects were selected as the largest 40 objects from the real-world size dataset, and small objects were selected as the smallest 40 objects from the same dataset. In addition, the same objects in a silhouette or texture were also used to evaluate the effect of shape and texture on the real-world size representation in the occipitotemporal cortex. Each run consisted of six conditions (i.e., big origin, small origin, big silhouette, small silhouette, big texture, and small texture). Each run lasted 320 s, which included four block sets. Each block set lasted 60 s, consisting of three conditions during which all 40 images were shown for each condition. Each image was presented for 200 ms, followed by a 300ms fixation, consisting of a 500 ms trial. The position of each object on the screen was slightly jittered to increase the attention of the participant. The first two block sets presented images from all of the six conditions without repetition, and the last two block sets presented images in palindrome. Five fixation periods of 16s intervened between each block set. Participants were instructed to pay attention to the images and to press a button when a red frame appeared around an object, which appeared twice per condition randomly.

*Preprocessing*. Anatomical and functional data were preprocessed using fMRIPrep (version 20.2.0)[55]. Preprocessing included skull stripping, slice-time correction, co-registration to T1w with boundary-based registration cost-function, correction for head-motion and susceptibility distortion, and temporal high-pass filtering (128 s cut-off). All structural and functional images were projected into a 32k_fs_LR space[56] using the ciftify toolbox[57]. The functional images were spatially smoothed with a 4 mm FWHM kernel.

*Univariate analyses*. First-level statistical analyses were performed for functional images of each participant in each run using the general linear model (GLM) from the HCP Pipelines. Second-level statistical analyses were separately performed on the activation maps generated from the first-level analysis.

To define the size-sensitive ROI, a whole-brain group analysis across participants was conducted based on the second-level statistical maps of the even runs (i.e., run 2, 4, 6, 8) using the Permutation Analysis of Linear Models (PALM)[58]. A contrast was performed at an uncorrected threshold of p < 0.005 to identify the regions selectively active to original big objects versus original small objects. The identified regions from the origin condition were used as a mask of regions sensitive to real-world size in the following analyses.

Activation magnitude was measured as Cohen's *d* with the formula,

$$d = \frac{\beta}{\sqrt{\text{dof} \times \text{var}(\beta)}} \tag{13}$$

where β is the contrast of parameter estimation (COPE), and dof is the degree of freedom. Activation magnitude in the pre-defined ROIs was extracted from odd runs (i.e., run 1, 3, 5, 7) for origin, silhouette and texture conditions of each participant, respectively.

*Multivariate analyses*. Multivariate analyses used data from odd runs (i.e., run 1, 3, 5, 7). We assessed whether the multivoxel patterns of different conditions (i.e., origin, silhouette, and texture) in the mask of the real-world size were sufficient to classify the size category (i.e., big or small) of the object being viewed. The classification was performed on a support vector machine (SVM) with a linear kernel using the leave-one-out cross-validation (LOOCV) across participants, and its accuracy was evaluated as the averaged accuracies from all runs of the LOOCV. The significance of classification accuracy was evaluated with a null distribution, which was built by classifications after permuting the pooled activations from the original conditions 10,000 times. Significance (*p* < 0.05) was achieved when classification accuracy was larger than 0.75. Searchlight analysis was also performed to

test for coding of real-world size in the whole brain. The same procedure of classification was evaluated in small spherical ROIs (radius 10 mm) centered on each vertex of the brain in turn. This generated three whole-brain maps corresponding to original objects, silhouettes, and textures. The searchlight maps were spatially smoothed using a Gaussian smoothing kernel with FWHM as 2mm.

**Statistics and reproducibility**. Statistical analyses were conducted using the statistical computing programming language Python (version 3.8.0). Results were visualized with Python package Matplotlib (https://matplotlib.org). Replicated analyses were performed in multiple DCNNs, including two VGG networks, two ResNet networks and one Inception network (Please see Supplementary Materials for replications). All visual stimuli used in this study were downloaded from open datasets. The fMRI experiment replicated the previous finding[8] of the medial-to-lateral arrangement of a big-to-small map separated by the MFS when the original objects were shown to the participants.

**Reporting summary**. Further information on research design is available in the Nature Research Reporting Summary linked to this article.

## Data availability

All data underlying our study and necessary to reproduce our results are available on Github: https://github.com/helloTC/RealWorldSizeAxis. The single-object version of the ImageNet ILSVRC2012 dataset is available on ScienceDB: https://www.scidb.cn/en/doi/10.57760/sciencedb.01674. Other datasets presented in this study can be found in online repositories, the names of each repository and the download location can be found in the article.

## Code availability

All code underlying our study and necessary to reproduce our results are available on Github: https://github.com/helloTC/RealWorldSizeAxis.

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

## Acknowledgements

This study was funded by the National Natural Science Foundation of China (31872786 and 31861143039), Shuimu Tsinghua Scholar Program (T.H.), Tsinghua University Guoqiang Institute (2020GQG1016), Tsinghua University Qiyuan Laboratory, and Beijing Academy of Artificial Intelligence (BAAI).

## Author contributions

T.H., Y.S., and J.L. designed the research and wrote the manuscript. T.H. performed experiments and analyzed data. All authors contributed to the article and approved the submitted version.

## Competing interests

The authors declare no competing interests.
