## [Peer Review File · Communications Biology]

Reviewers' comments:

Reviewer #1 (Remarks to the Author):

The authors explore the continuous dimension of real-world object size in deep neural networks trained on object recognition, testing it on a number of DNNs trained with different stimulus sets and architectures to explore the robustness of this representation. And they conduct a fMRI experiment with big and small objects in original, silhouette and texturized formats. I find the exploration of real-world size factor as a property of DNN feature spaces to be an interesting and important question, and the finding that it loads fairly selectively on PC2 to be surprising. However, the manuscript left me conceptually confused in many places, and I feel that it is not well situated in the previous literature in an attempt to be novel, which undermines its interesting connection to previous work. I think there fMRI imaging analyses are incomplete, and not at the same level as the modeling work.

>>Does the manuscript have technical or conceptual flaws that should prohibit its publication? If so, please provide details.

- The abstract, introductory framing, and claims make theoretical/conceptual assertions that I do not think are founded. For example, in the abstract: "Our mind can represent various objects from the physical world metaphorically into an abstract and complex high-dimensional object space, with a finite number of orthogonal axes encoding critical object features." Why do you assume these have to be orthogonal? I do not believe this is a correct assertion, e.g. orientation information is represented with overlapping tuning curves which are not orthogonal to each other. I think a finite orthogonal basis is an interesting idea, and a point worthy of theoretical discussion but not a pre-condition for a relevant feature of object representation by any means. Is there even evidence that DNNs are using finite orthogonal features to represent objects? I think not! (the orthogonality you're finding in the feature space is a direct property of your choice use PCA, but it is not as if the PC2 is not an explicit feature encoding in the DNN... rather this is just a way to probe the major factors in this high-d space. If the DNN was using orthogonal independent factors to represent objects, you'd have as many PCs as you had units. The point is, this theoretical line is strongly held throughout the abstract, intro, and results, and undermines my ability to focus on the results and contributions of the work.)

- I find the motivation of the work to be conceptually confusing and not sufficiently situated in the literature on high-level object recognition in the visual system.

- For example, the opening paragraph motivate the problem based almost entirely on the Bao 2020 science paper, and motivates their own work subsequently with an example of a flaw of the Bao account: "[a] tennis balls and the earth are both inanimate and stubby, but they belong to two object categories." I find this example reveals some conceptual confusion of the Bao et al, account. So, the relationship between Bao's proposal and the real-world size argument is not clearly presented in the set up.

- In the abstract: the motivation is: "little is known about what features serve as axes of the object space to critically affect object recognition". However, this is a very rich area of research, with many people working on this and there are many proposals. For example, Long et al., 2018 argues for mid-level features, organized by categorical properties of animacy and real-world object size (see also Konkle & Caramazza, 2011; and Konkle & Oliva, 2011). Much further work has been done thinking about the animacy dimension, and how it is related. Further, the recent work of Bao makes proposals about the axes! So this set up feels like it does not respect the prior work that this (interesting) project is built on. In general I think you are working very hard to find weaknesses in prior work in order to be novel and situate your work. However, I think your work would be stronger and better situated if it acknowledges the strengths and conceptual positions of the empirical work. And, then the relationship between size and Bao's axes could be more clearly situated. As it is now, I cannot actually tell if you think your findings are consistent, inconsistent, or unrelated to Bao's claims.

- I find the "necessity" analysis confusing—you took out variance from the feature space and the top-1 accuracy dropped by a few percent. Would this happen with any axis? Is there any way you can remove an axis of the feature space and *not* impact the top-1 accuracy? I found it difficult to understand the implications/alternatives, and wouldn't you expect a bigger drop maybe if this was such an important axis? Or something more like object category confusions errors that were not as size-related as when the size-axis was in there, perhaps?

fMRI analyses: I could not easily follow these analyses, e.g. I do not understand figure 5: what ROI are we looking in here? a big object preferring region? I expected to also see activations in a small-object preferring region? It might also be worth discussing how you reconcile your findings with the work of Tim Andrews and David Coggan who argue for textural representations throughout the ventral visual stream.

>> Are the conclusions original? If not, please provide relevant references.

- The broad conclusion, that real-world size is a key axis of object representation, has been offered previously (Konkle & Oliva, 2011; Konkle & Caramazza, 2013) but it has not been tested directly in DNNS). The title needs more appropriate scoping add "in object-recognition artificial neural networks."

- For example, the last sentence of the abstract: "our study provided the first evidence supporting the feature of objects' real-world size as an axis of object space, and devised a novel paradigm for future exploring the structure of object space." This conclusion needs more appropriate scoping: this is not the first evidence of object size as an axis of the human mind or human brain's representation of objects! But it is appropriate to claim that this is the first demonstration that the dimension of real-world size is a major factor in the feature spaces of deep neural networks (typically as the 2nd principle component), providing computational support for the empirical observations about the relevance of this factor in human perceptual and neural representations of objects. What is your novel paradigm for future explorations of object space?

- The authors also claim that the human brain "only humans only used the shape information, but not the texture information, to represent objects' real-world size in the ventral occipitotemporal cortex of the brain." I was unable to assess this claim given their data presentation. But more generally, so much is known about this topic, with many studies working to triangulate the underlying features of object responses. I refer you to Long et al., 2018, as an example of how local-texture information can recapitulate the object size and animacy organization (without the silhouette!). I actually think the work would be a bit stronger without the brain data (as they focus on only big vs small while the whole rest of the paper is about the real-world size logarithmic axis; they are not even asking about the dimension but instead digging into the feature relationship between real-world object size and texture and shape; and in truth the analyses are not at currently at the standard level of fMRI published research on visual object representation.)

>> Do you feel that the results presented are of immediate relevance for people in your own discipline or for a broader audience?

• The question is interesting: "Here we asked whether the feature of objects' real-world size constructed an axis of object space with deep convolutional neural networks (DCNNs)." This topic has never been explored as systematically as in this work. However, as stated above, I think motivating this work by the prior proposals that object size is a large-scale organizing factor of human ventral visual stream would be a nice way to acknowledge this influence on the work.

Other comments:

- Why was layer conv4 selected as the main layer to show? Bao used FC6. Given the strong motivation

contrasting with Bao's account, I was surprised a different layer was used. This also made it difficult to assess w.r.t their claims and your claims.

- I found the opening of the results difficult to understand without reading the methods (e.g. you could explain how you operationalized real-world object size (rank 1-8), and extracted activations (e.g. summing over convs, before the relu). Relatedly, sometimes you use rank and sometimes you use natural logarithm of size, and I couldn't quite understand the motivation for why both were used rather than one or the other

- I think the result that object size is encoding almost entirely/selectively in PC2 to be a very interesting result! But, I do not think the assertion this is a necessary criteria for an object axis is what makes it interesting... Rather that it is surprising that it is so selective and another control dimension that doesn't load onto a PC so selectively would help highlight this as a surprising/important result. Does this selective loading hold for all the layers or just conv4? Bao's analysis (Sup Fig 9) used big and small image stimuli and it seems like these vary along PC1. Perhaps reporting layer-wise results and doing a direct replication of Bao's graph would help clarify the issue..

- Conceptual clarity on "object space" as a term is needed (e.g. line 171). There is the theoretical hypothesis that there is a unified object space supporting encoding of all objects in the human/monkey brain (c.f. bao et al., 2020; Konkle & Oliva, 2013). Then, there are DNNs, whose layers operationalize different candidate object feature spaces, which may or may not be a perfect reflection of the hypothesized object space, or a perfect match with the brain measures (e.g. What about the object space of different layers?) It is difficult to keep track of the results, which are about the properties of DNN feature spaces, and the potential implications for the representational space of objects in the mind and for the response structure of the brain.

- Ln 177: I would like more detail about single object alexnet. Were all the objects presented at the same visual size or variable visual sizes during training? I think these image set training manipulations are interesting and valuable additions.

- I know that was a lot of feedback, but I think there are interesting results in here. Figuring out how to situate them in a way that clarifies what inferences we can make about what machine vision systems learn from the input is useful. They have feature information related to the dimensions of real-world object size without that even being a primary factor of the task! Nor do they manipulate or interact with them. That means those factors, which have been thought to play an influencing role in the brain, may be ben in the input statistics without requiring visuo-motor interactions. In any case, thinking along these lines I think might be a more productive framing on what we can learn about the human mind and brain from these findings. Good luck!

Reviewer #2 (Remarks to the Author):

This paper finds strong support for the idea that real-world object size is a robustly identifiable axis of object encoding in several different CNN architectures. It uses exploration of the model to find features that lead to real-world size estimation in human fMRI. My comments are mainly minor.

I don't understand how this encoding is independent, if all animals includes both big and small
"Finally, size information is apparently encoded independently from animacy, and they 69
together form a tripartite organizational schema (i.e., big objects, all animals and 70
small objects)"

I don't think the r value of the match to the ideal observer is reported for the scrambled case in Fig 4.

"The fMRI experiment on humans revealed that humans only used the shape information of objects to infer objects' real-world size," I think "only" is strong here, as there is still an unexplained gap between shape only and full performance, and there are other things besides the texture technique used by the authors that could've been tested.

The authors should reference and engage more with previous studies showing texture bias in CNNs (e.g. "Deep convolutional networks do not classify based on global object shape" and "The Origins and Prevalence of Texture Bias in Convolutional Neural Networks") to acknowledge this finding isn't new and contextualize theirs within this framework

line 334: "uttermost" does not make sense here

Given the reliance on shape, I'd like to hear the authors thoughts on how the visual system determines real world size of objects with the same shape, like the toy and real car example they discuss.

"We averaged responses from the convolutional layers 513 within each channel, resulting in a response pattern of 256 channels for each image. " - the 256 is presumably the number of channels which varies by layer. Is this specifically for layer 4?

I don't understand equation 1. What exactly are i and j ?

Reviewer #3 (Remarks to the Author):

Summary:

The authors of this paper want to 1) test whether an object's real-world size is an axis of object space and 2) look at what variables contribute to the real-world size axis. They use a combination of deep neural network modeling, behavioral experimentation, and fMRI to address these questions. Overall the manuscript is interesting, with clever methods. However I have some feedback that could make the answers to the above questions more convincing. I also believe there are a few errors in arguments posed by the authors, which should be explained.

Feedback on Introduction:

-I'm a fan of the object space hypothesis and think it's well-described in the literature. My one issue is that Talia Konkle's work (Long and Konkle duo of papers in 2016 (behavior) and 2018 (fMRI)) seem to indicate that real-world size can be well-explained from curvature. The authors of the current paper state that animacy and curvature are two axes of object-space and their paper explores the possibility that real-world size is a third. But according to their independence hypothesis, real-world size cannot possibly be a third since it correlates with curvature.

-On a similar note, later on in the intro, the authors describe Talia's work on tripartite organization, which is good evidence for independent axes of real-world size and animacy. Thus, hasn't the independence criterion been taken care of? What more does this study add as far as independence that Konkle & Caramazza 2013 didn't?

Figure 1 and 2:

-What would happen if authors redid Figure 1 with curvature rather than real-world size? More specifically, the authors could create an Ideal observer RDM based on some objective curvature stat (check out Yue et al. 2014/2020) and a human behavior RDM based on boxy/curvy ratings.

-It seems arbitrary to choose AlexNet (and specifically Conv4 layer) as the main architecture/layer focus of this paper. Although the authors tested other architectures and layers, this was not done for all figures. For instance, Conv3's RDM has a similar correlation to the ideal observer RDM than Conv4? What happens when you rerun ALL analyses with this layer? Or a combined Conv4/Conv3? Or the WHOLE Alexnet? Do results hold? Although authors attempted to test other architectures, this is not nearly to the scale that it needs to be for me to be sure this result is robust.

-I think that there could have been a different way to choose the architectures. Why not choose the 10 highest networks based on brainscore? Their rationale for choosing the architectures they did come across as post-hoc explanations based on architectures (more specifically architecture/network combinations) which "worked". What about models with recurrence? What about generative models? Are there shared architectural motifs in models that do work?

-Did the image-set used to construct the RDM in Fig 1 include animals? From the images shown on the left of Fig 1A, it seems that the answer was no. But from the description, it seems as if you used KonkLab's image set, which does include animals. This needs to be made clear. Not including animals in the RDM may create stronger real-world size distinctions, based on the fact that objects are likely grouped into real-world size based on affordance/manipulability, and that real-world size operates differently in animals as opposed to objects.

-I liked the PCA-based "lesion" approach. However, I would argue against the notion that this procedure worked for other architecture/layers. In Fig S4, the DI isn't always significant across different architecture/layers. The effect is a lot weaker. I'm nervous about the robustness of these results.

-For curiosity's sake, is there any idea what PC1 was? Relatedly to this point, is PC2 special? It seems to be the PC with high dropout index across all architecture/layers tested. Authors don't seem to mention this, or elaborate.

-On the topic of Figure 2, the bar-graph in the right panel should be explained more. Were the categories in which accuracy was impaired the same across all architecture/networks? This would be compelling, but this is not clear in the text. It would be even more interesting if there was something about the categories in which accuracy was impaired that tied them more closely to real-world size. Perhaps you could show exemplars of those categories to human participants and see if they are less accurate at big/small discrimination.

-What was the significance test for Figure 2C,left?

-The natural log relationship in 2B is interesting and well-founded, especially from Talia's work (as the authors in this paper cite). Even more of a tie-in/more detailed explanation would be appreciated, as the result from Talia's work is totally different in details.

-An overarching question about Figure 2 is: does this actually test the independence criterion? Wouldn't you have to actually test whether real-world size is independent of other object axes to truly satisfy the criterion?

The rest of the figures:

-For Figure 3, I'd like to see this exercise done for other DCNN's. Why wasn't it? Also, there's no explanation of why animate/artifact was the categorization task. Does this approach work for other categorization tasks?

-In Fig 4C, was natural log the best fit, compared to other relationships? The authors should clarify

this.

-Authors should consider using texforms for Fig 4. These are stimuli that preserve texture/form but make stimuli unrecognizable at the exemplar level. It could be that some combination of texture/form actually comprises real-world size, which would be in line with Long & Konkle 2018.

-The texture images used are very clearly less recognizable than the silhouette images, which may be driving the effects in Fig 5.

-What specifically about shape is driving the real-world size effect? Could it be curvature? If its curvature, this harkens back to my earlier comments about independence.

-The motivation and inferences made from Figure 6 are unclear. DCNNs seem to use texture and shape to represent real-world size. Human brains seem to only use texture. Do the authors want to convey that the DCNN's don't actually *need* texture, although it can help? This seems like a weak addition and doesn't merit a separate figure in my opinion.

Writing clarity:

-Authors say that "after identifying the feature of objects' real-world size as an axis of object space, a more interesting question is how the DCNN acquired [real-world size] information"... this is odd considering what they did in Fig 1 and 2 was really the motivation behind the paper, as well as what was focused on in the introduction, as well as the title of the manuscript!

-One small point is that the behavioral paradigm used in Figure 1 should be at least alluded to in the text. It's unclear from just reading the manuscript what the human behavior RDM originates from.

-Some spelling grammatical issues, nothing major

RE: Real-world size of objects serves as an axis of object space (COMMSBIO-21-3130)

Response to Reviewers:

We would like to thank the reviewers for their appreciation of our study and thoughtful comments. Responding to the reviewers has helped to improve the manuscript, and has been a rewarding experience for us. Reviewers' comments have been addressed in a point-by-point manner. The comments are **in bold**, and our responses are immediately below.

Reviewer 1

Q1: The abstract, introductory framing, and claims make theoretical/conceptual assertions that I do not think are founded. For example, in the abstract: "Our mind can represent various objects from the physical world metaphorically into an abstract and complex high-dimensional object space, with a finite number of orthogonal axes encoding critical object features." Why do you assume these have to be orthogonal? I do not believe this is a correct assertion, e.g. orientation information is represented with overlapping tuning curves which are not orthogonal to each other. I think a finite orthogonal basis is an interesting idea, and a point worthy of theoretical discussion but not a pre-condition for a relevant feature of object representation by any means. Is there even evidence that DNNS are using finite orthogonal features to represent objects? I think not! (the orthogonality you're finding in the feature space is a direct property of your choice use PCA, but it is not as if the PC2 is not an explicit feature encoding in the DNN... rather this is just a way to probe the major factors in this high-d space. If the DNN was using orthogonal independent factors to represent objects, you'd

have as many PCs as you had units. The point is, this theoretical line is strongly held throughout the abstract, intro, and results, and undermines my ability to focus on the results and contributions of the work.)

R1: We agree with the reviewer that the hypothesis on orthogonality is largely assumed in the literature, but not directly and soundly verified. In our study, we found that objects' size was encoded in the PC2 alone, supporting the hypothesis of orthogonality among axes of the object space. However, we also agree that this does not necessarily suggest the explicit use of this orthogonal feature in DCNNs. In addition, we also agree that the finite number of axes of object space is also assumed with no theoretical proof.

In revision, we have removed these two hypotheses (i.e., orthogonality and the finite number of axes) on object space in the abstract and introduction, and only in the discussion we briefly discuss the orthogonality of axes (Line 358-363): “..., *we found that the size axis, like axes for animacy and curvature¹⁻⁴, was statistically orthogonal to other axes. This is important, because from the perspective of the resource rationality^{5,6}, it is uneconomical if object features are used to construct object space are redundant and correlated. Therefore, orthogonality among axes of object space is likely to reduce the redundancy of visual details effectively^{7,8}.*”

Q2: I find the motivation of the work to be conceptually confusing and not sufficiently situated in the literature on high-level object recognition in the visual system. For example, the opening paragraph motivate the problem based almost entirely on the Bao 2020 science paper, and motivates their own work subsequently with an example of a flaw of the bao account: “[a] tennis balls and the earth are both inanimate and stubby, but they belong to two object categories.” I find this example reveals some conceptual confusion of

the Bao et al, account. So, the relationship between Bao’s proposal and the real-world size argument is not clearly presented in the set up.

In the abstract: the motivation is: “little is known about what features serve as axes of the object space to critically affect object recognition”. However, this is a very rich area of research, with many people working on this and there are many proposals. For example, Long et al., 2018 argues for mid-level features, organized by categorical properties of animacy and real-world object size (see also Konkle & Caramazza, 2011; and Konkle & Oliva, 2011). Much further work has been done thinking about the animacy dimension, and how it is related. Further, the recent work of Bao makes proposals about the axes! So this set up feels like it does not respect the prior work that this (interesting) project is built on. In general I think you are working very hard to find weaknesses in prior work in order to be novel and situate your work. However, I think your work would be stronger and better situated if it acknowledges the strengths and conceptual positions of the empirical work. And, then the relationship between size and bao’s axes could be more clearly situated. As it is now, I cannot actually tell if you think your findings are consistent, inconsistent, or unrelated to Bao’s claims.

R2: We are sorry for the confusion. We have no intention of emphasizing the novelty of this study by ignoring previous work on the representation of objects’ size. Instead, we considered “orthogonality” as a criterion for axes of object space, which has not been thoroughly tested in previous studies. With the removal of this criterion as the reviewer suggested, we revised the introduction extensively, and all related studies have been properly cited.

Abstract (Line 18-24): *“Among object features identified in previous neurophysiological and fMRI studies that may serve as the axes, objects’ real-world size is of particular interest because of its ecological property for*

understanding objects' affordance. Here we used deep convolutional neural networks (DCNNs), which enable direct manipulation of visual experience and units' activation, to explore how objects' real-world size was extracted to construct the axis of object space."

Introduction (Line 46-52): *"Thus, object recognition is considered a computational problem of finding multiple axes to build a simplified division surface to separate different objects represented by features projected from these axes^{9,10}. Previous neurophysiological studies suggest that this hypothetical object space is implemented in human and non-human primates' inferotemporal cortex, such as real-world size (big versus small)^{2,11-14}, animacy (animate versus inanimate)^{1,2,11,12,15} and curvature (spiky versus stubby)^{1,3,4}."*

Q3: I find the "necessity" analysis confusing—you took out variance from the feature space and the top-1 accuracy dropped by a few percent. Would this happen with any axis? Is there any way you can remove an axis of the feature space and *not* impact the top-1 accuracy? I found it difficult understand the implications/alternatives, and wouldn't you expect a bigger drop maybe if this was such an important axis? Or something more like object category confusions errors that were not as size-related as when the size-axis was in there, perhaps?

R3: We agree with the reviewer that the effect size is small, though the decrease in accuracy was significant. The reason we used the criterion "necessity" is to emphasize the involvement of this axis in object recognition, but "necessity" is apparently overstated. In revision, we have removed this criterion throughout the text, and only reported the involvement of the size axis in object recognition.

Result (Line 148-157): *"As the definition of object space that axes are constructed to facilitate object recognition, we then examined the role of the size axis in object*

recognition with an ablation analysis that is not applicable in biological systems. That is, we removed Conv4's response variance aligned to this axis to examine whether AlexNet's performance on object recognition was impaired. Specifically, with the residue responses after regressing out PC2, AlexNet's Top-1 accuracy of recognizing the ImageNet validation images was slightly but significantly decreased from 52.6% to 48.5% ($p < 0.001$) in general (Fig 2C, left), indicating that the size axis contributes to object recognition. Similar results were found in other DCNNs (Fig S5C) as well."

Q4: fMRI analyses: I could not easily follow these analyses, e.g. I do not understand figure 5: what ROI are we looking in here? a big object preferring region? I expected to also see activations in a small-object preferring region?

R4: We are sorry for the confusion. The ROI in Figure 5A was defined by the contrast of big objects versus small objects in the object condition of the odd runs (i.e., run 1, 3, 5, 7). Regions in red showed preference for big objects, whereas regions in blue for small objects. The pre-defined ROI was then used to extract brain responses for all three conditions of even runs (i.e., run 2, 4, 6, 8), which are shown in Figure 5B and C.

Details about the definition of the ROI were added in the Method (Line 635-646): *"To define the size-sensitive ROI, a whole-brain group analysis across participants was conducted based on the second-level statistical maps of the even runs (i.e., run 2, 4, 6, 8) using the Permutation Analysis of Linear Models (PALM)¹⁶. A contrast was performed at an uncorrected threshold of $p < 0.005$ to identify the regions selectively active to big objects versus small objects. The identified regions from the object condition were used as a size-sensitive ROI in following analyses.*

Activation magnitude was measured as Cohen's d with the formula,

$$d = \frac{\beta}{\sqrt{\text{dof} \times \text{var}(\beta)}} \quad (12)$$

where β is the contrast of parameter estimation (COPE), and dof is the degree of freedom. Activation magnitude in the pre-defined ROI was extracted from odd runs (i.e., run 1, 3, 5, 7) for object, silhouette and texture conditions of each participant, respectively.”

Q5: It might also be worth discussing how you reconcile your findings with the work of Tim Andrews and David Coggan who argue for textural representations throughout the ventral visual stream.

R5: Coggan et al. (2016, 2019) used both locally-scrambled (intact shape information but unrecognizable) and globally-scrambled images (no shape information, similar to our texture stimuli) to examine neural representation in the ventral visual pathway, and they found the representation for the locally-scrambled images were similar to that of the original ones, but distinctly different from that of the globally-scrambled images. This finding is consistent with ours that shape information was necessary to differentiate objects’ size.

In revision, we have discussed Coggan et al.’s findings (Line 258-268): “*A thorough whole-brain searchlight analysis did not find any region capable of distinguishing objects’ size based on texture information (Fig 5D). This finding is in line with previous studies that when shape information is removed from the texforms¹⁷ or when images are globally scrambled^{18,19}, the neural activation in the ventral visual pathway is no longer similar to that of their original ones. Taken together, the human brain apparently relied extensively more on shape information than texture information to infer object’s size.*”

Q6: The broad conclusion, that real-world size is a key axis of object representation, has been offered previously (Konkle & Oliva, 2011; Konkle & Caramazza, 2013) but it has not been tested directly in DNNS). The title needs more appropriate scoping add “in object-recognition artificial neural networks.”

- For example, the last sentence of the abstract: “our study provided the first evidence supporting the feature of objects’ real-world size as an axis of object space, and devised a novel paradigm for future exploring the structure of object space.” This conclusion needs more appropriate scoping: this is not the first evidence of object size as an axis of the human mind or human brain’s representation of objects! But it is appropriate to claim that this is the first demonstration that the dimension of real-world size is major factor in the feature spaces of deep neural networks (typically as the 2nd principle component), providing computational support for the empirical observations about the relevance of this factor in human perceptual and neural representations of objects. What is your novel paradigm for future explorations of object space?

R6: We are sorry for the confusion. In the original manuscript, we proposed three criteria to define axes of object space, which is much stricter than the previous ones. In revision, all three criteria have been removed, and we have therefore acknowledged that objects’ size as an axis of object space has already been proposed in previous studies (for details, please see R1&R2).

Q7: The authors also claim that the human brain “only humans only used the shape information, but not the texture information, to represent objects’ real-world size in the ventral occipitotemporal cortex of the brain.” I was unable to assess this claim given their data presentation. But more generally, so much is

known about this topic, with many studies working to triangulate the underlying features of object responses. I refer you to Long et al., 2018, as an example of how local-texture information can recapitulate the object size and animacy organization (without the silhouette!).

R7: Our finding did not contradict Long et al.'s. In their study, texform stimuli actually consist of shape information, and therefore it is not surprising that texform stimuli can recapitulate objects' size. In contrast, in our study, stimuli in the texture condition contain no shape information at all. In revision, we have clarified the difference in stimuli, and discussed Long et al.'s study.

Results (Line 240-243): *“The finding that either shape or texture of objects alone was sufficient to infer objects' size in DCNNs echoes neuroimaging studies on humans that texform stimuli, which preserve both texture and shape information but are not recognizable, can successfully recapitulate objects' size^{17,20,21} .”*

And (Line 263-266): *“This finding is in line with previous studies that when shape information is removed from the texforms¹⁷ or when images are globally scrambled^{18,19}, the neural activation in the ventral visual pathway is no longer similar to that of their original ones.”*

Q8: I actually think the work would be a bit stronger without the brain data (as they focus on only big vs small while the whole rest of the paper is about the real-world size logarithmic axis; they are not even asking about the dimension but instead digging into the featureal relationship between real-world object size and texture and shape; and in truth the analyses are not at currently at the standard level of fMRI published research on visual object representation.)

R8: The fMRI experiment was inspired by the analysis of the DCNN's response profile that both shape and text information was used for the representation of

objects' real-world size. It is natural to ask whether human uses these two types of information to infer objects' size because human's ventral visual cortex and DCNNs are similar but not identical. Therefore, we believe the fMRI experiment is necessary.

Because the fMRI experiment is not the major point of the study, we tried to make it concise, which unfortunately led to confusion to some extent. In revision, we have included a detailed description of data analysis (please see R4), explained the difference in stimuli (texture stimuli in our study versus texform stimuli, locally-scrambled stimuli) and discussed related references (please see R5 and R7). We hope that the revision makes the fMRI study more related to our study on DCNNs.

Q9: The question is interesting: “Here we asked whether the feature of objects’ real-world size constructed an axis of object space with deep convolutional neural networks (DCNNs).” This topic has never been explored as systematically as in this work. However, as stated above, I think motivating this work by the prior proposals that object size is a large-scale organizing factor of human ventral visual stream would be a nice way to acknowledge this influence on the work.

R7: We thank the reviewer for the suggestion, and we completely agree. In revision, we have extensively revised both the abstract and introduction to using previous neuroimaging studies on humans to motivate this study (for details, please see R2).

Q10: Why was layer conv4 selected as the main layer to show? Bao used FC6. Given the strong motivation contrasting with Bao’s account, I was surprised a

different layer was used. This also made it difficult to assess w.r.t their claims and your claims.

R10: As shown in Supplementary Figure 1, the FC2 layer, which was called FC6 in Bao et al.'s study, also showed strong correspondence to the ideal observer ($r=0.87$, $p<0.001$). Unlike Bao et al.'s study, we introduced the ideal observer as a baseline, which allows us to objectively locate the layer with the highest sensitivity to objects' size; therefore, we chose the Conv4 layer showing the highest correspondence ($r = 0.94$) as the layer of interest for further analyses, rather than the FC2 layer used in Bao et al.'s study where no baseline was used.

We now clarify our reasons for choosing the Conv4 in the Results (Line 110-113): *“Furthermore, the similarity was not restricted to the Conv4 layer; instead, all convolution layers except the Conv1 layer showed sensitivity to objects' size (Fig S1), with the Conv4 layer showing the highest correspondence.”*

Q11: I found the opening of the results difficult to understand without reading the methods (e.g. you could explain how you operationalized real-world object size (rank 1-8), and extracted activations (e.g. summing over convs, before the relu). Relatedly, sometimes you use rank and sometimes you use natural logarithm of size, and I couldn't quite understand the motivation for why both were used rather than one or the other.

R11: Thanks for the suggestion. In revision, we moved a part of the method section to the result section to improve the legibility.

Results: (Line 98-100) *“An ideal observer was constructed as a baseline to represent the size relation among objects, which was used to measure how closely the representation of objects' size matched the ground truth (Fig 1A).”*

(Line 104-110) *“In addition, to examine the similarity between AlexNet's responses and human's subjective experience on objects' size, we also measured*

human's judgment on objects' size where participants were instructed to choose a larger object from object pairs randomly sampled from the same dataset. We found human's subjective experience on object size was highly similar to AlexNet's responses to it (Fig 1C; $r = 0.94$, $p < 0.001$), suggesting at least a weak equivalence between DCNN and human in representing the size feature of objects."

(Line 123) "To examine whether the size feature served as an axis of object space, ..."

(Line 136-141) "A great challenge to encoding objects' size is that the size varies greatly (e.g., airplanes are 2-3 orders of magnitude larger than basketball) and the size of daily objects is in a heavy tail distribution, with the concentration mainly in the range of centimeters to meters. To examine how this axis efficiently encodes objects' real-world size, we tested a variety of encoding schemes, such as linear, power, and logarithm functions."

(Line 148-150) "As the definition of object space that axes are constructed to facilitate object recognition, we then examined the role of the size axis in object recognition with an ablation analysis that is not applicable in biological systems."

Q12: I think the result that object size is encoding almost entirely/selectively in PC2 to be a very interesting result! But, I do not think the assertion this is a necessary criteria for an object axis is what makes it interesting... Rather that it is surprising that it is so selective and another control dimension that doesn't load onto a PC so selectively would help highlight this as a surprising/important result. Does this selective loading hold for all the layers or just conv4? Bao's analysis (Sup Fig 9) used big and small image stimuli and it seems like these vary along PC1. Perhaps reporting layer-wise results and doing a direct replication of Bao's graph would help clarify the issue.

R12: Indeed, it is surprising that object size was selectively encoded in PC2, suggesting that this feature is represented orthogonally to other features in object space. To examine whether the selectivity was present in other layers as well, we performed a layer-wise analysis (Reviewer Figure 1, New Supplemental Figure 4).

Reviewer Figure 1 (New Fig S4). Layer-wise analysis on the selectivity of PC2 in encoding objects' real-world size. The encoding of objects' size started from the Conv3 layer to the FC3 layer. Importantly, this feature was selectively encoded in PC2 in all these layers.

As shown in Reviewer Figure 1, objects' size was selectively encoded in PC2 starting from the Conv3 layer. This new analysis is now included in the revised text.

A close inspection of the FC layers suggests that objects' size might be encoded in PC1 to some extent; however, the loading was not significant (all $ps > 0.05$). To further verify the selectivity, we followed the analysis in Bao et al.'s study as instructed. We downloaded a new dataset from Konkle's lab (<https://konklab.fas.harvard.edu/ImageSets/BigSmallObjects.zip>), where objects were binarily classified into big and small categories. We fed all images into a pre-

trained AlexNet, and extracted the PC1 and PC2 in each layer. Our results confirmed that PC2 is the only axis that encoded objects' size (Reviewer Figure 2).

Reviewer Figure 2. Differences in objects' size were distinguishable along PC2, but not along PC1 in AlexNet starting from the Conv3 layer.

Q13: Conceptual clarity on “object space” as a term is needed (e.g. line 171).

There is the theoretical hypothesis that there is a unified object space supporting encoding of all objects in the human/monkey brain (c.f. bao et al., 2020; Konkle & Oliva, 2013). Then, there are DNNs, whos layers operationalize different candidate object feature spaces, which may or may not be a perfect reflection of the hypothesized object space, or a perfect match with the brain measures (e.g. What about the object space of different layers?) It is difficult to keep track of the results, which are about the properties of DNN feature spaces, and the potential implications for the representational space of objects in the mind and for the response structure of the brain.

R13: This is a great point! Unfortunately, our study is unable to address this question since we only studied one axis of object space in DCNNs. Our guess is that object spaces of different layers, if they do exist, are not complete, because successfully object recognition is only achieved at the final layer of DCNNs. This

is actually similar to what is hypothesized in the human/monkey brain, where complete object space may be finally constructed at the higher level of the visual cortex. In other words, at different layers, object features for axes of object space are extracted and represented (e.g., objects' real-world size in our study), and only at the final layers are the object features assembled into complete object space for object recognition.

In the revised discussion, we have clarified this point (Line 375-377): *“First, in this study we only focused on objects' size, and it is interesting to know how this feature works collaboratively with other features to construct object space to represent objects as a whole.”*

Q14: Ln 177: I would like more detail about single object alexnet. Were all the objects presented at the same visual size or variable visual sizes during training? I think these image set training manipulations are interesting and valuable additions.

R14: The single object AlexNet was used to rule out the possibility that objects' size was derived from the comparison in retina size among objects presented in the training image set. Therefore, the images that were used to train the single object AlexNet were not adjusted to the same visual size; instead, the visual size of each image was kept as it was in the original one. We appreciate that the reviewer considered this control analysis valuable and we will make the single object dataset publicly available for future studies in the communities.

This is now clarified in the Method section: (Line 482-483) *“Note that the retinal size of objects remained unchanged with only the background was removed from the original images.”*

The single-object version of ImageNet ILSVRC2012 was available online:
(Line 820-822) “*The single-object version of the ImageNet ILSVRC2012 dataset is available on ScienceDB: <https://www.scidb.cn/en/doi/10.57760/sciencedb.01674>.*”

Reviewer 2.

Q1: I don't understand how this encoding is independent, if all animals includes both big and small "Finally, size information is apparently encoded independently from animacy, and they together form a tripartite organizational schema (i.e., big objects, all animals and small objects)"

R1: We are sorry for the confusion. We agree that the assumption of independence among axes of object space is not conclusive (also pointed out by reviewer #1, please see R1 to Reviewer #1), which is now removed from the text. In revision, we have modified it in the Introduction section (Line 67-69): “..., and along with animacy, artefacts’ real-world size forms a tripartite organizational schema (i.e., big artefacts, animals, and small artefacts) in the ventral occipitotemporal cortex² and medial temporal lobe¹. ”

Q2: I dont think the r value of the match to the ideal observer is reported for the scrambled case in Fig 4.

R2: Thanks for pointing it out. This missing information ($R^2 = 0.01$) is now included in Fig 4.

Q3: "The fMRI experiment on humans revealed that humans only used the shape information of objects to infer objects’ real-world size," I think "only" is strong here, as there is still an unexplained gap between shape only and full performance, and there are other things besides the texture technique used by the authors that could've been tested.

R3: We agree. Here we mainly focused on the contrast between texture and shape, and we have no intention to state that shape is the only factor. This confusion is now clarified in the Results section (Line 301-304): “Taken together, the visual experience of correct texture information of objects was not necessary to

infer objects' size, implying a strong equivalence between DCNNs and human in representing objects' real-world size."

Q4: The authors should reference and engage more with previous studies showing texture bias in CNNs (e.g. "Deep convolutional networks do not classify based on global object shape" and "The Origins and Prevalence of Texture Bias in Convolutional Neural Networks") to acknowledge this finding isn't new and contextualize theirs within this framework.

R4: We are sorry for missing these important studies. In revision, these studies have been properly cited and discussed (Line 283-285): *"The difference between human and DCNNs in relying on texture information to infer objects' size may reflect the fact that DCNNs are heavily biased by objects' texture^{22,23}, which is primarily originated from the training data²⁴."*

Q5: line 334: "uttermost" does not make sense here.

R5: Sorry for the wording, which has been corrected in revision.

Q6: Given the reliance on shape, I'd like to hear the authors thoughts on how the visual system determines real world size of objects with the same shape, like the toy and real car example they discuss.

R6: This is an interesting point! Indeed, as many illusions demonstrate, the shape is not the only factor that defines objects' size. High-level factors, such as context, also help resolve ambiguity in sizes (e.g., toy cars versus real cars). For example, toy cars are unlikely seen on highways, and therefore any car on the highway is considered a big object with high confidence. In revision, we explicitly explain this in the Discussion (Line 378-382): *"Second, objects' shape was found as a key factor to infer objects' size; however, the underlying mechanism is largely*

unknown, as objects of similar size may vary greatly in shape (e.g., Eiffel tower versus Ford-class aircraft carrier). Future studies need to explore what specifically about shape information drives objects' real-world size."

Q7: "We averaged responses from the convolutional layers within each channel, resulting in a response pattern of 256 channels for each image. " - the 256 is presumably the number of channels which varies by layer. Is this specifically for layer 4?

R7: Yes, the 256 channels here are from the Conv4 layer, which is specified in the Methods (Line 510-512): "*We averaged responses from convolutional layers within each channel, resulting in a response pattern of 256 channels in Conv4 for each image.*"

Q8: I don't understand equation 1. What exactly are i and j?

R8: Sorry for the confusion, which is clarified in the Methods (Line 523): "*where i and j are indicators of size ranks ranging from 1 to 8.*"

Reviewer 3.

Q1: I'm a fan of the object space hypothesis and think it's well-described in the literature. My one issue is that Talia Konkle's work (Long and Konkle duo of papers in 2016 (behavior) and 2018 (fMRI)) seem to indicate that real-world size can be well-explained from curvature. The authors of the current paper state that animacy and curvature are two axes of object-space and their paper explores the possibility that real-world size is a third. But according to their independence hypothesis, real-world size cannot possibly be a third since it correlates with curvature.

R1: In our study, we showed that shape information is important to infer objects' size. Because curvature is an important component of shape, our finding does not argue against the relation between the role of curvature in representing objects' size.

To further quantitatively examine the relation between curvature and objects' size as the reviewer suggested, we first measured the object's curvature and then examined its relation with its size.

Reviewer Figure 3. Curvature representation in AlexNet. Curvature was measured by aspect ratio, corresponding to objects' protrusion. Similar to real-world size of objects, we divided all images into 8 curvature ranks. Image numbers were kept the same for each size rank. (A) The ideal observer for curvature. (B) The representational similarity matrix of each layer to different curvatures. Noted that the Conv4 layer did not show high correspondence to the ideal observer, suggested curvature was independent of the real-world size of objects. (C) The low similarity between curvature and real-world size of objects.

Specifically, we used the object's aspect ratio to index its curvature suggested by Bao et al.'s study (2020), and performed the same procedure to generate the ideal observer of curvature (Reviewer Figure 3A) and the RDM of each layer in AlexNet. We found that the correspondence of the RDM to the ideal observer reached a peak in the FC1 layer (Reviewer Figure 3B). In contrast, the highest correspondence for objects' size was found in Conv4, arguing against a direct link between curvature and objects' size. To quantitatively examine their relation, we directly calculated the correlation between objects' curvature and size, and found no significant relation ($R^2=0.04$, $p>0.05$; Reviewer Figure 3C). Taken together, there is apparently no direct link between curvature and size. Instead, curvature information may be integrated with other components into the shape that in turn infers size information.

Having said this, we did not argue that curvature is represented in an axis orthogonal to the axis for objects' size, because as Reviewer #1 pointed out, the independence among axes of object space is assumed but not soundly tested in the literature. In revision we have toned down the statement by removing the criterion of independence (please see R1 to Reviewer #1).

Q2: On a similar note, later on in the intro, the authors describe Talia's work on tripartite organization, which is good evidence for independent axes of

real-world size and animacy. Thus, hasn't the independence criterion been taken care of? What more does this study add as far as independence that Konkle & Caramazza 2013 didn't?

R2: The tripartite organization of objects' size and animacy in the ventral temporal cortex does not necessarily suggest independence among axes, because their relationship was not directly tested. In revision, we have removed the criterion of independence, and only discussed it at the end of the manuscript (please see R1 to Reviewer #1 and R1 to Reviewer #2).

Q3: What would happen if authors redid Figure 1 with curvature rather than real-world size? More specifically, the authors could create an Ideal observer RDM based on some objective curvature stat (check out Yue et al. 2014/2020) and a human behavior RDM based on boxy/curvy ratings.

R3: Please see R1.

Q4: It seems arbitrary to choose AlexNet (and specifically Conv4 layer) as the main architecture/layer focus of this paper. Although the authors tested other architectures and layers, this was not done for all figures. For instance, Conv3's RDM has a similar correlation to the ideal observer RDM than Conv4? What happens when you rerun ALL analyses with this layer? Or a combined Conv4/Conv3? Or the WHOLE Alexnet? Do results hold? Although authors attempted to test other architectures, this is not nearly to the scale that it needs to be for me to be sure this result is robust.

R4: We re-run all analyses for different layers of the AlexNet, and found that objects' size was represented in all layers starting from Conv3. For details, please see R12 to Reviewer #1.

Q5: I think that there could have been a different way to choose the architectures. Why not choose the 10 highest networks based on brainscore? Their rationale for choosing the architectures they did come across as post-hoc explanations based on architectures (more specifically architecture/network combinations) which “worked”. What about models with recurrence? What about generative models? Are there shared architectural motifs in models that do work?

R5: The reason we chose DCNNs is based on enormous previous studies (including studies from our lab) that have shown functional similarities between DCNNs and primate/human ventral visual systems. Indeed, in our study, we showed that both DCNNs and the human ventral visual pathway used shape information to infer objects’ size. Based on this rationale, we expect that neural networks that have shown functional similarity to the human ventral visual cortex in previous studies (e.g., neural networks with high brain scores) shall encode objects’ size in a similar way.

Q6: Did the image-set used to construct the RDM in Fig 1 include animals? From the images shown on the left of Fig 1A, it seems that the answer was no. But from the description, it seems as if you used KonkLab’s image set, which does include animals. This needs to be made clear. Not including animals in the RDM may create stronger real-world size distinctions, based on the fact that objects are likely grouped into real-world size based on affordance/manipulability, and that real-world size operates differently in animals as opposed to objects.

R6: We are sorry for the confusion. The size dataset includes animals (under the label of living things), which consists of clown, cow, fish, German Shepard, kitten, toddler and traffic cop (Reviewer Figure 4A). It is an interesting point that animals

may affect the representation of objects' size. To test this intuition, we removed all animal images and then compared the RDM of Conv4 with the original RDM. The result suggests a little effect of animals in constructing the axis of objects' size, as it was highly similar to that with animals ($r=0.99$) (Reviewer Figure 4B). This finding suggests that the factor of affordance/manipulability is not necessary for representing objects' size.

Reviewer Figure 4 (A) Exemplars of living things from the size dataset. (B) After removing all images of living things, the RDM showed high correspondence to the original RSM in the Conv4 layer.

Q7: I liked the PCA-based “lesion” approach. However, I would argue against the notion that this procedure worked for other architecture/layers. In Fig S4, the DI isn't always significant across different architecture/layers. The effect is a lot weaker. I'm nervous about the robustness of these results.

Reviewer Figure 5 (New Fig S5). The real-world size axis emerged in object space in DCNNs with different architectures. (A) The size axis could be found in five different DCNNs, including two VGG networks, two ResNet networks, and one Inception network. The size axis from the ResNet34 was not significant while had a tendency. (B) Logarithm correspondence between the PC2 and real-world size of objects of the five DCNNs. (C) The top-1 accuracy decreased when the variance of the size axis was removed for all DCNNs. *, $p < 0.05$; ***, $p < 0.001$.

R7: We agree the effect was small, and the criterion of necessity is apparently overstated. In revision, we have removed this criterion (please see R3 to Reviewer #1). Besides, we also performed layer-wise analyses with this lesion approach. During this process, we realized the formula to calculate DI was not appropriate for all layers, because if R was a small value in some layers, the DI became a large negative value. Therefore, we revised the formula from $DI = 1 - \frac{r_i}{R}$ to $DI = Z(R) - Z(r_i)$. Where $Z(*)$ is the Fisher z-transformation. We replicated the result with this new formula, and the new DI was significant in all DCNNs tested (please see Reviewer Figure 5).

Q8: For curiosity's sake, is there any idea what PC1 was? Relatedly to this point, is PC2 special? It seems to be the PC with high dropout index across all architecture/layers tested. Authors don't seem to mention this, or elaborate.

R8: Because the scope of this study is on objects' size, we did not report the feature encoded by PC1, which might be sensitive to animacy (i.e., animals vs. artefacts) of objects. This conjecture comes from our test with the AnimacySize dataset (<https://konklab.fas.harvard.edu/ImageSets/AnimacySize.zip>), which contains objects of Big Animals, Big Artefacts, Small Animals and Small Artefacts (Reviewer Figure 6). Our result suggests a clear distinction between objects' size and animacy, where PC1 encoded animacy and PC2 represented objects' size. Further studies are needed to elucidate object features encoded in other PCs.

Reviewer Figure 6. (A) Exemplars of objects in the animacy-size dataset. (B) PC1 values of different categories, which were derived from activation of Conv4 in AlexNet. Error bar: standard deviation.

Q9: On the topic of Figure 2, the bar-graph in the right panel should be explained more. (1) Were the categories in which accuracy was impaired the same across all architecture/networks? This would be compelling, but this is not clear in the text. (2) It would be even more interesting if there was something about the categories in which accuracy was impaired that tied them more closely to real-world size. (3) Perhaps you could show exemplars of those categories to human participants and see if they are less accurate at big/small discrimination.

Reviewer Figure 7. (A) Consistency of category ranks of impairment in recognition between DCNNs. (B) Categories that were mostly affected by the removal of the size axis of AlexNet.

R9: Following the reviewer's suggestion, we measured the decrease in recognition accuracy of all object categories for each neural network tested, and then measured the consistency of rank order across categories (Reviewer Figure 7A). We found that ranks of the impaired categories in AlexNet, VGG11, VGG13 and ResNet18 shared intermediate consistency with each other ($r_s = 0.3\sim 0.6$), with low correspondence between ResNet34 and InceptionV3 ($r < 0.10$). We also explored object categories that were mostly affected after removing the size axis in the AlexNet, and found the top-5 categories that were most likely impaired: scoreboard, library, container ship, keeshond and fire engine, respectively (Reviewer Figure 7B). There is no clear clue on the relation between the degree of impairment and the real-world size (please also see R7, and R3 to Reviewer #1). It is possible that objects' size works collaboratively with other object features to recognize objects as a whole.

We have clarified this point in the Discussion: (Line 375-377) “..., in this study we only focused on objects' size, and it is interesting to know how this feature works collaboratively with other features to construct object space to represent objects as a whole.”

Q10: What was the significance test for Figure 2C, left?

R10: The significance of impairment in recognition accuracy was evaluated by comparing it with a null distribution, which was generated from an untrained DCNN with a repetition of 1,000 times. Specifically, a baseline object space was constructed from an untrained DCNN by decomposing Conv4's activations to 50,000 ImageNet ILSVRC validation images using PCA. The variance that aligned

to the second axis of the baseline object space was then regressed out from Conv4's activation in response to images from the size dataset. The changes in the accuracy of object recognition after the regression was compared to the original one without regression. In this way, a null distribution was constructed. The actual decrease in accuracy against this distribution was used as a measure of significance.

In revision, this is now clarified in Methods (Line 581-586): “*Similar to the measurement of DI, we first removed the variance of each PC from the original responses in Conv4, and then fed these responses into higher layers to get the Top-1 accuracy of the AlexNet for each category. Significance was tested by comparing it with a null distribution, which was generated from baseline object spaces originating from an untrained AlexNet with a repetition of 1000 times.*”

Q11: The natural log relationship in 2B is interesting and well-founded, especially from Talia's work (as the authors in this paper cite). Even more of a tie-in/more detailed explanation would be appreciated, as the result from Talia's work is totally different in details.

R11: We thank the reviewer for pointing this out. The logarithm property of the axis for objects' size is likely for coding compression. In revision, we have discussed it in detail.

Result (Line 136-146): “*A great challenge to encoding objects' size is that the size varies greatly (e.g., airplanes are 2-3 orders of magnitude larger than basketball) and the size of daily objects is in a heavy tail distribution, with the concentration mainly in the range of centimeters to meters. To examine how this axis efficiently encodes objects' real-world size, we tested a variety of encoding schemes, such as linear, power, and logarithm functions. Among all functions examined, the best function that maps the physical world (i.e., real-world size) to the representational space (i.e., values of PC2) was the natural logarithm scale*

($R^2 = 0.48$, $p < 0.001$, Fig 2B). That is, the stimulus-representation mapping follows the Weber-Fechner law, suggesting that AlexNet compresses large physical intensity ranges (i.e., objects' real-world size) into smaller response ranges of units.”

Discussion (Line 364-371): “... , the mapping from objects' real-world size to the size represented in the size axis was not linear; instead, the coding scheme adopted by the size axis followed the Weber-Fechner Law that the scale for large size was compressed exponentially, similar to the nonlinear coding strategy adopted by human²⁵. The advantage of the compressed scaling is likely to enlarge coding space to avoid explosions in the number of preferred neurons needed²⁶ and to increase the tuning range of neurons²⁷ so that they can efficiently encode objects with sizes differing in order of magnitude.”

Q12: An overarching question about Figure 2 is: does this actually test the independence criterion? Wouldn't you have to actually test whether real-world size is independent of other object axes to truly satisfy the criterion?

R12: We used PCA to extract axes of object space, which mathematically guaranteed the independence among axes. Besides, objects' size was only encoded in the second axis, indicating that this feature was encoded independently from the rest of the features.

Having said this, we realized that the independence of the axes in the object space is largely assumed in the literature but not validated. Therefore, in revision we have removed this criterion (for details, please see R1 to Reviewer #1).

Q13-1: For Figure 3, I'd like to see this exercise done for other DCNN's. Why wasn't it?

R13-1: To verify this result in other DCNNs, we trained a single-object VGG11 with the same procedure as the single-object AlexNet (Reviewer Figure 8). The top-1 accuracy of the VGG11 is 59.8%. We replicated the results from the single-object AlexNet, suggested by a close correspondence of response to the ideal observer ($r=0.95$, $p<0.001$; Layer: Features 11), and an axis specifically encoding the real-world size of objects ($DI=0.46$, $p<0.05$; Bonferroni Correction) with natural logarithm relationship ($R^2=0.43$, $p<0.001$). Again, this new analysis suggests that the object co-occurrence had little effect on the representation of objects' real-world size.

Reviewer Figure 8. The single-object VGG11 represents objects' real-world size.

Q13-2 Also, there's no explanation of why animate/artifact was the categorization task. Does this approach work for other categorization tasks?

R13-2: Previous studies from our lab have demonstrated that task demands may affect the representation of DCNNs^{28,29}. Accordingly, here we tested whether task demands affected the representation of objects' size.

To further illustrate the reliability of this finding as the reviewer suggested, we re-trained an AlexNet (AlexNet-Cate19) from scratch to classify objects into 19 super-ordinate categories, including fungus, fish, bird, amphibian, reptile, canine, primate, feline, ungulate, invertebrate, conveyance, device, container, equipment, implement, furnishing, covering, commodity, and structure suggested by the

WordNet³⁰. The stimulus and parameters used for the training were identical to the AlexNet-Cate2. The top-1 accuracy of AlexNet-Cate19 for object recognition is 68.7%. In this new analysis, we replicated the results from the AlexNet-Cate2 (Reviewer Figure 9). That is, the task demand had little effect on the representation of objects' real-world size ($r=0.96$, $p<0.001$), which was encoded by the same axis (DI=1.30, $p<0.05$; Bonferroni correction) with natural logarithm function ($R^2=0.46$, $p<0.001$).

Reviewer Figure 9. The effect of task demands on the representation of objects' size. From top to bottom, three DCNNs with the same architecture (i.e., AlexNet) were trained to classify objects at different levels of categorization. DCNNs were trained to categorize a car as (A) artefact (the highest superordinate level), or (B) conveyance (superordinate level), or (C) cab (basic level). Regardless of the levels of categorization, the axis for objects' size was found ubiquitously.

Q14: In Fig 4C, was natural log the best fit, compared to other relationships? The authors should clarify this.

R14: Yes, the natural logarithm was the best fit among all relationships tested. This is now clarified in the revised text (Line 141-143): “Among all functions examined,

the best function that maps the physical world (i.e., real-world size) to the representational space (i.e., values of PC2) was the natural logarithm scale ($R^2 = 0.48, p < 0.001$, Fig 2B)."

Q15: Authors should consider using texforms for Fig 4. These are stimuli that preserve texture/form but make stimuli unrecognizable at the exemplar level. It could be that some combination of texture/form actually comprises real-world size, which would be in line with Long & Konkle 2018.

R15: This is a great point! In our original study, we did not test texforms because we planned to directly contrast shape with texture in representing objects' size. We found that in DCNNs both types of information were sufficient for inferring the real-world size of objects. Therefore, it is reasonable to expect that objects' size can also be inferred from texforms, which preserve both texture and shape information.

Indeed, we confirmed this intuition by testing the texforms from Long et al.'s study (2018, <https://github.com/brialorelle/texformgen>). That is, the RDM of Conv4's responses to the texforms showed high correspondence to the ideal observer based on the original objects ($r=0.92, p<0.001$), which was also encoded in the second axis of the object space ($R^2=0.34, p<0.001$) (Reviewer Figure 10).

Reviewer Figure 10. Objects' real-worlds can also be recovered from texforms.

In revision, we have clarified the motivation of using silhouette and texture images rather than texforms in our study (Line 217-222): “*To thoroughly decouple the effect of shape and texture in representing objects’ size, we adopted silhouette images, which preserve overall shape information with no texture information, and texture images, which preserve only texture information. Shuffle images served as a control condition, where neither shape nor texture information was preserved.*”

Q16: The texture images used are very clearly less recognizable than the silhouette images, which may be driving the effects in Fig 5.

R16: We agree with the reviewer that the texture images are less recognizable than the silhouette images. However, recognizability is apparently not a prerequisite for recovering objects’ size from images. First, as demonstrated in our study, DCNNs were able to infer objects’ size from the texture images. Second, as shown in previous studies^{17,20,21}, humans are able to infer objects’ size from texforms, which are not recognizable either. Therefore, the size information is preserved in the texture, but humans apparently did not use texture to infer objects’ size.

Q17: What specifically about shape is driving the real-world size effect? Could it be curvature? If its curvature, this harkens back to my earlier comments about independence.

R17: This is an interesting question, and frankly we do not know. First, the curvature is unlikely the main reason behind the shape for inferring objects’ size (please see R3). Second, even if it was, the question still remains (i.e., what specifically about curvature is driving the real-world size effect?). In revision, we acknowledge this unaddressed issue (Line 378-382): “*..., objects’ shape was found as a key factor to infer objects’ size; however, the underlying mechanism is largely unknown, as objects of similar size may vary greatly in shape (e.g., Eiffel tower*

versus Ford-class aircraft carrier). Future studies need to explore what specifically about shape information drives objects' real-world size."

Q18: The motivation and inferences made from Figure 6 are unclear. DCNNs seem to use texture and shape to represent real-world size. Human brains seem to only use texture. Do the authors want to convey that the DCNN's don't actually *need* texture, although it can help? This seems like a weak addition and doesn't merit a separate figure in my opinion.

R18: We are sorry for the confusion. In our study, we found that humans used shape information, rather than texture information, to infer objects' size, whereas DCNNs used both types of information. To reconcile the difference, we tested whether the texture information is not only sufficient but also necessary for DCNNs to infer objects' size. This is the reason we used the stylized AlexNet, which is less biased by texture than the original AlexNet, and the answer is negative that texture information was apparently not necessary for DCNNs to infer objects' size.

In revision, we have clarified the motivation (Line 283-290): *"The difference between human and DCNNs in relying on texture information to infer objects' size may reflect the fact that DCNNs are heavily biased by objects' texture^{22,23}, which is primarily originated from the training data²⁴. Therefore, it is not surprising that DCNNs used texture information to infer objects' size. However, the finding from humans raised an interesting question: is texture information necessary for DCNNs to infer objects' size, given that texture information is unstable in the natural environment, largely affected by climates, air, or illumination?"*

Q19: Authors say that "after identifying the feature of objects' real-world size as an axis of object space, a more interesting question is how the DCNN

acquired [real-world size] information”... this is odd considering what they did in Fig 1 and 2 was really the motivation behind the paper, as well as what was focused on in the introduction, as well as the title of the manuscript!

R19: We are sorry for the confusion. Here we tried to identify factors that were used by DCNNs to infer objects’ size. In revision, the motivation for these further analyses is clarified (Line 170-174): *“As expected, the size axis was absent in an untrained AlexNet (Fig S3), showing the necessity of stimulus experience, rather than DCNNs’ architecture, in constructing the size axis. To further explore factors that contributed to the construction of the size axis, we first examined the context that provides the relative difference in retinal size among objects.”*

Q20: One small point is that the behavioral paradigm used in Figure 1 should be at least alluded to in the text. It’s unclear from just reading the manuscript what the human behavior RDM originates from.

R20: The introduction to the behavioral experiment is now added in Results (Line 104-107): *“In addition, to examine the similarity between DCNNs and humans in representing objects’ size, we also measured human’s subjective judgment on objects’ size where participants were instructed to choose a large object from object pairs randomly sampled from the same object dataset.”*

R21: Some spelling grammatical issues, nothing major

R21: We are sorry for the grammatical mistakes. In revision, we have corrected them thoroughly.

Reference

1. Bao, P., She, L., McGill, M. & Tsao, D. A map of object space in primate inferotemporal cortex. *Nature* **583**, 103–108 (2020).
2. Konkle, T. & Caramazza, A. Tripartite organization of the ventral stream by animacy and object size. *Journal of Neuroscience* **33**, 10235–10242 (2013).
3. Yue, X., Pourladian, I. S., Tootell, R. B. & Ungerleider, L. G. Curvature-processing network in macaque visual cortex. *Proceedings of the National Academy of Sciences* **111**, E3467–E3475 (2014).
4. Yue, X., Robert, S. & Ungerleider, L. G. Curvature processing in human visual cortical areas. *NeuroImage* **222**, 117295 (2020).
5. Gershman, S. J., Horvitz, E. J. & Tenenbaum, J. B. Computational rationality: A converging paradigm for intelligence in brains, minds, and machines. *Science* **349**, 273–278 (2015).
6. Lieder, F. & Griffiths, T. L. Resource-rational analysis: Understanding human cognition as the optimal use of limited computational resources. *Behavioral and Brain Sciences* **43**, (2020).
7. Barak, O., Rigotti, M. & Fusi, S. The sparseness of mixed selectivity neurons controls the generalization–discrimination trade-off. *Journal of Neuroscience* **33**, 3844–3856 (2013).

8. Vidal, Y., Viviani, E., Zoccolan, D. & Crepaldi, D. A general-purpose mechanism of visual feature association in visual word identification and beyond. *Current Biology* **31**, 1261–1267 (2021).
9. DiCarlo, J. J., Zoccolan, D. & Rust, N. C. How does the brain solve visual object recognition? *Neuron* **73**, 415–434 (2012).
10. Shepard, R. N. Toward a universal law of generalization for psychological science. *Science* **237**, 1317–1323 (1987).
11. Blumenthal, A., Stojanoski, B., Martin, C. B., Cusack, R. & Köhler, S. Animacy and real-world size shape object representations in the human medial temporal lobes. *Human brain mapping* **39**, 3779–3792 (2018).
12. Grill-Spector, K. & Weiner, K. S. The functional architecture of the ventral temporal cortex and its role in categorization. *Nature Reviews Neuroscience* **15**, 536–548 (2014).
13. Julian, J. B., Ryan, J. & Epstein, R. A. Coding of object size and object category in human visual cortex. *Cerebral Cortex* **27**, 3095–3109 (2017).
14. Konkle, T. & Oliva, A. A real-world size organization of object responses in occipitotemporal cortex. *Neuron* **74**, 1114–1124 (2012).
15. Sha, L. *et al.* The animacy continuum in the human ventral vision pathway. *Journal of cognitive neuroscience* **27**, 665–678 (2015).

16. Winkler, A. M., Ridgway, G. R., Webster, M. A., Smith, S. M. & Nichols, T. E. Permutation inference for the general linear model. *Neuroimage* **92**, 381–397 (2014).
17. Long, B., Konkle, T., Cohen, M. A. & Alvarez, G. A. Mid-level perceptual features distinguish objects of different real-world sizes. *Journal of Experimental Psychology: General* **145**, 95 (2016).
18. Coggan, D. D., Liu, W., Baker, D. H. & Andrews, T. J. Category-selective patterns of neural response in the ventral visual pathway in the absence of categorical information. *Neuroimage* **135**, 107–114 (2016).
19. Coggan, D. D., Baker, D. H. & Andrews, T. J. Selectivity for mid-level properties of faces and places in the fusiform face area and parahippocampal place area. *European Journal of Neuroscience* **49**, 1587–1596 (2019).
20. Long, B., Yu, C.-P. & Konkle, T. Mid-level visual features underlie the high-level categorical organization of the ventral stream. *Proceedings of the National Academy of Sciences* **115**, E9015–E9024 (2018).
21. Long, B. & Konkle, T. Mid-level features are sufficient to drive the animacy and object size organization of the ventral stream. *Journal of Vision* **17**, 575–575 (2017).

22. Baker, N., Lu, H., Erlikhman, G. & Kellman, P. J. Deep convolutional networks do not classify based on global object shape. *PLoS computational biology* **14**, e1006613 (2018).
23. Geirhos, R. *et al.* ImageNet-trained CNNs are biased towards texture; increasing shape bias improves accuracy and robustness. *arXiv preprint arXiv:1811.12231* (2018).
24. Hermann, K., Chen, T. & Kornblith, S. The origins and prevalence of texture bias in convolutional neural networks. *Advances in Neural Information Processing Systems* **33**, 19000–19015 (2020).
25. Konkle, T. & Oliva, A. Canonical visual size for real-world objects. *Journal of Experimental Psychology: human perception and performance* **37**, 23 (2011).
26. Dehaene, S. & Changeux, J.-P. Development of elementary numerical abilities: A neuronal model. *Journal of cognitive neuroscience* **5**, 390–407 (1993).
27. Dayan, P. & Abbott, L. F. *Theoretical neuroscience: computational and mathematical modeling of neural systems*. (MIT press, 2005).
28. Huang, T., Zhen, Z. & Liu, J. Semantic relatedness emerges in deep convolutional neural networks designed for object recognition. *Frontiers in computational neuroscience* **15**, 16 (2021).

29. Song, Y., Qu, Y., Xu, S. & Liu, J. Implementation-independent representation for deep convolutional neural networks and humans in processing faces. *Frontiers in computational neuroscience* **14**, (2020).
30. Miller, G. A. WordNet: a lexical database for English. *Communications of The ACM* **38**, 39–41 (1995).

Reviewers' comments:

Reviewer #1 (Remarks to the Author):

Overall I found the paper much improved. It was clear the authors did quite a lot of work for this revision, and I appreciate their due diligence! The way the results are written also now make it more easy for me to see and understand the contributions of the work.

My remaining comments relate to the new way this work is theoretically situated in the intro (and echoed in the general discussion)--which I think is better than before, but still not quite accurate, and thus confusing. Currently, the setup is something like: "object size is only relevant for vision-for-action which is a dorsal stream kind of task... so why would it be a factor in the ventral stream? Surprisingly, we find real-world size in deep nets, which are supposed to be a model of ventral stream!" e.g. a direct quote: " This finding is surprising because DCNNs is thought to simulate the human ventral visual pathway that is dedicated to vision for perception, whereas numerous fMRI and neuropsychological studies have demonstrated the function of vision for action is performed in the dorsal visual pathway".

But, critically it is already known and documented that real-world size is a relevant factor for the ventral stream! So, the current introduction is not really situating the relevance and theoretical importance of your contributions well. Below I offer a way that, to me, clarifies the theoretical role your work plays in understanding the representation of objects in both biological and artificial systems, by more directly situating it in what is known and unknown about size and ventral stream representation. It is not important to me that you use this exact set up, per se. If you want to go for a different set up that also acknowledges what's known about the ventral stream, that's great too! (Also, feel free to use any sentences/phrases from what I've written below, if you want to).

"The real-world size of objects has been shown to be an important factor in structure of responses along the ventral visual stream (Konkle & Oliva, 2011). There are at least two non-exclusive reasons why this real-world size distinction emerges that have been proposed. One factor is related to action-based demands-- that is, there are likely different output demands needed for small and big objects, and these featural distinction in the ventral stream may be guided by those action requirements, e.g. potentially realized through differential connectivity with dorsal-manipulation vs medial-navigation networks (c.f. Konkle & Caramazza, 2016, Cerebral Cortex). However, the other factor that might underlie the real-world size distinction in the ventral stream is that objects of different sizes have distinct visual image statistics (e.g. Long et al., 2016 JPEG). Indeed, the real-world size organization of the ventral stream is even evoked by unrecognizable images which only preserve mid-level localized texture and coarse form information, indicating a visuo-statistical representational basis, rather than an abstract representation of real-world size. However, it is still not clear whether these featural distinctions along the human ventral stream emerge due to pressure from action requirements, or covariance in input statistics. Deep convolutional neural networks trained to do object categorization provide a new computational test-bed to explore these origin stories, because they don't have action-based pressures, but they excel at extracting relevant image statistics for categorization. To what degree is the dimension of real-world size part evident in the representational space of objects learned in deep neural networks?"

I hope my paragraph above might help give a sense of how I see this work as novelly adding to our advancing understanding of the ventral visual stream, and object representation more generally. Again, there are many possible set ups. I just want very much for you to get an on-ramp to your work that doesn't have to downplay what is already known, in order to bring novelty to your work. I think the work you've done is very important and interesting, in the context of what is known, and adds exciting new evidence to bear on the broad goal of understanding object representation.

Reviewer #3 (Remarks to the Author):

Hi,

Absolute improvement from the original. I like the smaller focus on orthogonality (based on Rev1's comments) and it seems narrower, clearer in scope. I also liked a clearer review of the literature to date.

That being said, still needs a little work.

Just to summarize, the authors are studying 1) whether object size is encoded in DNNs trained to recognize objects 2) what properties of an object influence this object size representation

~~

Comments on revised manuscript:

-Authors should bring up PC1 at some point in the manuscript. PC1 seems to be animacy based on response to my original review. Why wasn't this addressed? Have other studies showed animacy and real-word size encoding in DNN's?

-Why 2C? Is there any scenario we'd expect there not to be a drop in accuracy if you lesion a layer? This result seems obvious.

-What aspects of shape differentiate big vs small stimuli? This isn't addressed by authors. Curvature is definitely a candidate hypothesis that wasn't tested adequately by authors.

-The fMRI piece isn't central to the main work. The main questions they are testing are the one's I listed above. What's the motivation for including this here?

-If you want to include fMRI piece, I suggest a swapping of Fig 5 and 6. It definitely flows better as: "shape or texture sufficient for DNN RWS encoding ==> texture not necessary ==> VTC only uses shape info for RWS encoding

-I liked reviewer figure 9 (try many different categorization tasks) and think it can strengthen Fig 3D, as it is now, cab vs artifact seems arbitrary.

RE: Real-world size of objects serves as an axis of object space (COMMSBIO-21-3130B)

Response to Reviewers:

We would like to thank the reviewers for their appreciation of our revision and constructive comments on our manuscript. Reviewers' comments have been addressed in a point-by-point manner. The comments are **in bold**, and our responses are immediately below.

Reviewer 1

Q1: Overall I found the paper much improved. It was clear the authors did quite a lot of work for this revision, and I appreciate their due diligence! The way the results are written also now make it more easy for me to see and understand the contributions of the work.

My remaining comments relate to the new way this work is theoretically situated in the intro (and echoed in the general discussion)--which I think is better than before, but still not quite accurate, and thus confusing. Currently, the setup is something like: "object size is only relevant for vision-for-action which is a dorsal stream kind of task... so why would it be a factor in the ventral stream? Surprisingly, we find real-world size in deep nets, which are supposed to be a model of ventral stream!" e.g. a direct quote: " This finding is surprising because DCNNs is thought to simulate the human ventral visual pathway that is dedicated to vision for perception, whereas numerous fMRI and neuropsychological studies have demonstrated the function of vision for action is performed in the dorsal visual pathway".

But, critically it is already known and documented that real-world size is a relevant factor for the ventral stream! So, the current introduction is not really situating the relevance and theoretical importance of your contributions well. Below I offer a way that, to me, clarifies the theoretical role your work plays in understanding the representation of objects in both biological and artificial systems, by more directly situating it in what is known and unknown about size and ventral stream representation. It is not important to me that you use this exact set up, per se. If you want to go for a different set up that also acknowledges what's known about the ventral stream, that's great too! (Also, feel free to use any sentences/phrases from what I've written below, if you want to).

"The real-world size of objects has been shown to be an important factor in structure of responses along the ventral visual stream (Konkle & Oliva, 2011). There are at least two non-exclusive reasons why this real-world size distinction emerges that have been proposed. One factor is related to action-based demands-- that is, there are likely different output demands needed for small and big objects, and these featural distinction in the ventral stream may be guided by those action requirements, e.g. potentially realized through differential connectivity with dorsal-manipulation vs medial-navigation networks (c.f. Konkle & Caramazza, 2016, Cerebral Cortex). However, the other factor that might underlie the real-world size distinction in the ventral stream is that objects of different sizes have distinct visual image statistics (e.g. Long et al., 2016 JPEG). Indeed, the real-world size organization of the ventral stream is even evoked by unrecognizable images which only preserve mid-level localized texture and coarse form information, indicating a visuo-statistical representational basis, rather than an abstract representation of real-world size. However, it is still not clear whether these

featural distinctions along the human ventral stream emerge due to pressure from action requirements, or covariance in input statistics. Deep convolutional neural networks trained to do object categorization provide a new computational test-bed to explore these origin stories, because they don't have action-based pressures, but they excel at extracting relevant image statistics for categorization. To what degree is the dimension of real-world size part evident in the representational space of objects learned in deep neural networks?"

I hope my paragraph above might help give a sense of how I see this work as novelly adding to our advancing understanding of the ventral visual stream, and object representation more generally. Again, there are many possible set ups. I just want very much for you to get an on-ramp to your work that doesn't have to downplay what is already known, in order to bring novelty to your work. I think the work you've done is very important and interesting, in the context of what is known, and adds exciting new evidence to bear on the broad goal of understanding object representation.

R1: We deeply appreciate the reviewer's comments and especially the detailed example in writing that connects our study with previous ones! We also thank the reviewer's generous offer of using his/her words in our manuscript, which articulates the rationale of this study more precisely. In revision, we again extensively revised the Introduction and Discussion sections to strengthen the connection.

Introduction:

(Line 59-81) "*..., the functionality of objects' real-world size suggests at least three sources that may account for the development of size sensitivity in the VTC. The most evident source is that objects of different sizes likely have distinct mid-*

level perceptual properties (e.g., local corners, junctions, and contours) that are extracted at the early stages of visual processing, as in visual search the target object is detected faster when it differs in real-world size with the distractor objects¹ and the cortical region with size sensitivity is evoked by unrecognizable objects of different sizes that only preserve coarse form information².

On the other hand, the real-world size describes the scale of an object in natural environment, which implies the potential layout of the object³. For example, the size of whales suggests its co-occurrence with seas but not creeks. Therefore, the size sensitivity may derive from the context information that describes co-occurrence of multiple objects and their relations to the environment observed in daily life. Finally, the real-world size provides heuristic information for affordance⁴, which a specific real-world size of an object is associated with a set of specific actions, but not every action. Accordingly, the size sensitivity may be guided by action, which is potentially realized through differential connectivity with dorsal-manipulation versus medial-navigation networks⁵. Note that these three sources for the development of the size sensitivity are not mutually exclusive, because in daily life they are usually tightly intermingled and therefore hardly decoupled in conventional experiments to evaluate the contribution of each source independently.”

(Line 90-94) “..., a set of DCNNs are specifically designated for perception (i.e., object recognition) without top-down semantic modulation or action-based task demands; therefore, we can examine to what degree the axis of real-world size is evident in the object space constructed in the DCNNs that only analyze image statistics of objects.”

Discussion:

(Line 390-403) *“In this study, we focused on the feature of objects’ real-world size that not only facilitates object recognition (vision for perception) but also heuristically affects our action on objects (vision for action)⁶. We found that DCNNs solely designated for perception automatically encoded the feature of object’s size, immune to context and action-based task demands, implying that the perceptual analysis of objects’ shape was likely the main source for our brain to develop the sensitivity to objects’ size, to infer objects’ relation to the environment, and to find appropriate actions upon objects. In fact, neuropsychological studies on patients reveal that objects’ shape was used for actions on the objects, though the patients were not consciously aware of the objects⁷⁻⁹, and our fMRI experiment, along with previous studies^{1,2,10}, also showed that shape information alone (i.e., silhouettes) was sufficient to activate the size-sensitive regions in the VTC. Therefore, the perceptual analysis of objects’ shape seems a pre-requisite for action on the objects.”*

Reviewer 3

Q1: Authors should bring up PC1 at some point in the manuscript. PC1 seems to be animacy based on response to my original review. Why wasn't this addressed? Have other studies showed animacy and real-word size encoding in DNN's?

Reviewer Figure 1 (New Fig. 8 C, D). (A) The size axis (PC2) was not sensitive to the size of animals ($t=1.78$, $p=0.08$). (B) Instead, the feature of animacy was apparently encoded in the first principal component (PC1) of the object space, as the values of PC1 distinguished animacy from artefacts ($t=13.05$, $p<0.001$). ***: $p < 0.001$; n.s.: not significant.

R1: Yes, PC1 in the object space apparently encoded the feature of animacy in the DCNN. This is interesting; however, the systematic examination on the feature of animacy is beyond the scope of this study. Therefore, we have reported a brief analysis on the feature of animacy by examining its relation to the feature of objects' size in the revised text.

Results (Line 346-354): “Another well-established axis of object space is animacy (animate versus inanimate), which forms a tripartite organizational schema with objects' size (i.e., big artefacts, animals, and small artefacts)¹¹. Consistent with the observation in human, the size axis was not sensitive to the size

of animals (Fig 8C; $t=1.78$, $p=0.08$). Instead, the feature of animacy was apparently encoded in the first principal component (PC1) of the object space, as the values of PC1 distinguished animacy from artefacts (Fig 8D; $t=13.05$, $p<0.001$). In sum, the feature of objects' size was independent of the feature of curvature and animacy in the object space."

Methods (Line 559-564): *"To investigate the relationship between the size axis and objects' animacy, an animacy-size dataset was downloaded from <https://konklab.fas.harvard.edu/ImageSets/AnimacySize.zip>, which contains background-free objects of big animals, big artefacts, small animals, and small artefacts. For each type of objects, the dataset included 60 images, which consists of 240 images in total."*

Q2: Why 2C? Is there any scenario we'd expect there not to be a drop in accuracy if you lesion a layer? This result seems obvious.

R2: We are sorry for the confusion. This ablation analysis (i.e., the removal of variance aligned to the size axis) was included to connect with previous studies on human. In revision, this is now clarified (line 156-159): *"Behavioral studies on humans have revealed that the sensitivity to objects' size significantly facilitates object recognition¹. Accordingly, here we explicitly examined the role of the size axis in object recognition with an ablation analysis that is not applicable in biological systems."* Besides, the removal of variance itself does not necessarily impair networks' performance. For example, in our study, the removal of the variance of a pre-trained AlexNet that was aligned to the axis of the object space derived from an untrained AlexNet did not significantly impair the performance of the pre-trained AlexNet. In fact, this was used as a baseline (i.e., the condition of

No Size Axis) to measure the decrease in accuracy after the removal of variance aligned to the size axis (i.e., the condition of Origin) in Fig. 2C.

Q3: What aspects of shape differentiate big vs small stimuli? This isn't addressed by authors. Curvature is definitely a candidate hypothesis that wasn't tested adequately by authors.

R3: This is a great question. However, understanding exactly which mid-level perceptual properties embedded within shape separate big and small objects is beyond the scope of this study, because mid-level properties (e.g., corner) derive from combinations of simpler visual properties (two line segments in a particular geometric relation) and therefore the possible set of mid-level properties is unconstrained. In revision, we have included the analysis on the relation between the feature of curvature and the feature in the result section.

Results (Line 334-346): *“Both the fMRI and DCNN experiments suggest the critical role of shape in extracting the feature of objects' size. Among all types of shapes, curvature (spiky versus stubby) is most related because it is an axis of object space¹² and a mid-level stimulus property that may provide important information for objects' size (e.g., big objects are boxier but small objects curvier). Therefore, it is possible that curvature and objects' size may share the same axis of object space. To test this possibility, we measured objects' curvature by calculating objects' aspect ratio¹², with larger values indicating spiky objects and small values for stubby objects. We found that there was no significant correlation between curvature and real-world size of objects (Fig 8A; $R^2=0.01$, $p=0.41$). Besides, the loading of curvature on PC2 was small (Fig 8B; $R^2=0.001$, $p=0.72$), which was much smaller than the loading of objects' size ($R^2=0.48$), suggesting*

that the size axis unlikely relied on curvature as an important shape property to represent objects' size.”

Reviewer Figure 2 (New Fig 8. A, B). The independence of coding objects' size and curvature. Curvature was measured by aspect ratio, which showed no significant correlation with either objects' size (A) or PC2 (B).

Besides, In revised Discussion, we have also acknowledged the limitation of this study (Line 425-430): “..., objects' shape was found as a key factor to infer objects' size; however, in the study we only tested one mid-level perceptual property of curvature, and found that it had little contribution in representing objects' size. Future studies need to explore which mid-level properties, such as local corners, junctions, and contours, embedded within shape information, provide real-world size distinctions between objects.”

Q4: The fMRI piece isn't central to the main work. The main questions they are testing are the one's I listed above. What's the motivation for including this here? If you want to include fMRI piece, I suggest a swapping of Fig 5 and 6. It definitely flows better as: "shape or texture sufficient for DNN RWS encoding ==> texture not necessary ==> VTC only uses shape info for RWS encoding

R4: We agree that the fMRI experiment is not central to the main work; instead, it provides a comparison between DCNNs and human brain in understanding mid-level perceptual properties (i.e., shape versus texture) in representing objects' size. The result that human size-sensitive cortical regions were not evoked by texture alone led to a follow-up study on whether texture was necessary for constructing the size axis in DCNNs. In revision, this logic is now clarified.

Results (line 259-266): *“The finding that either shape or texture of objects alone was sufficient to infer objects' size in DCNNs echoes neuroimaging studies in human that texform stimuli, which preserve both texture and shape information but are not recognizable, can successfully recapitulate objects' size^{2,10}. However, behavioral studies on humans show that texture alone is not able to infer objects' size¹, which is apparently contrary to the role of texture observed in DCNNs. To further explore the role of texture on objects' size in humans, we asked whether the texture shown to AlexNet could activate cortical regions with size sensitivity in the VTC.”*

Q5: I liked reviewer figure 9 (try many different categorization tasks) and think it can strengthen Fig 3D, as it is now, cab vs artifact seems arbitrary.

R5: We are grateful that the reviewer appreciate this new analysis. In revision, we have included this analysis in the main text (new Fig 4.)

Results (Line 209-219): *‘To test top-down task demands on the representation of objects' size, we trained two new AlexNets with the same image datasets but to differentiate objects at a coarse level of living things versus artefacts (i.e., the AlexNet-Cate2) or at a superordinate level of 19 categories (i.e., the AlexNet-*

Cate19, see Methods for all superordinate categories; Reviewer Figure 3A). Again, the task demands had little effect on the representation of objects' size, with the best correspondence to the ideal observer in Conv4 (Reviewer Figure 3B; AlexNet-Cate2: $r = 0.95$, $p < 0.001$; AlexNet-Cate19: $r=0.96$, $p < 0.001$) and the same axis encoding objects' size (AlexNet-Cate2: $DI = 1.30$, $p < 0.05$, corrected; AlexNet-Cate19: $DI=1.30$, $p<0.05$, corrected) with a common logarithm function (AlexNet-Cate2: $R^2 = 0.37$, $p < 0.001$; AlexNet-Cate19: $R^2=0.46$, $p<0.001$) (Reviewer Figure 3C).

Reviewer Figure 3 (New Fig 4.) Factors of objects' task demands in inferring objects' real-world size. (A) Task demands. Three DCNNs with the same architecture (i.e., AlexNet) were trained to classify objects at different levels of categorization. DCNNs were trained to categorize a car as cab (basic level), conveyance (superordinate level), or artefact (coarse level). Note that the DCNN for the basic-level categorization is the same as the one used in the previous

experiments. (B) The RSM of Conv4's responses of the AlexNets with different task demands. From left to right: basic level, superordinate level, and coarse level. (C) The second axis of the object space also specifically encoded the real-world size with the mapping function of the common logarithm. S: small objects; B: big objects.

,

References

1. Long, B., Konkle, T., Cohen, M. A. & Alvarez, G. A. Mid-level perceptual features distinguish objects of different real-world sizes. *Journal of Experimental Psychology: General* **145**, 95 (2016).
2. Long, B., Yu, C.-P. & Konkle, T. Mid-level visual features underlie the high-level categorical organization of the ventral stream. *Proceedings of the National Academy of Sciences* **115**, E9015–E9024 (2018).
3. Julian, J. B., Ryan, J. & Epstein, R. A. Coding of object size and object category in human visual cortex. *Cerebral Cortex* **27**, 3095–3109 (2017).
4. Tucker, M. & Ellis, R. The potentiation of grasp types during visual object categorization. *Visual cognition* **8**, 769–800 (2001).
5. Konkle, T. & Caramazza, A. The large-scale organization of object-responsive cortex is reflected in resting-state network architecture. *Cerebral cortex* **27**, 4933–4945 (2017).
6. Vingerhoets, G., Vandamme, K. & Vercammen, A. Conceptual and physical object qualities contribute differently to motor affordances. *Brain and Cognition* **69**, 481–489 (2009).
7. Goodale, M. A., Westwood, D. A. & Milner, A. D. Two distinct modes of control for object-directed action. *Progress in brain research* **144**, 131–144 (2004).

8. Goodale, M. A. & Humphrey, G. K. The objects of action and perception. *Cognition* **67**, 181–207 (1998).
9. Milner, A. D. & Goodale, M. A. Two visual systems re-viewed. *Neuropsychologia* **46**, 774–785 (2008).
10. Long, B. & Konkle, T. Mid-level features are sufficient to drive the animacy and object size organization of the ventral stream. *Journal of Vision* **17**, 575–575 (2017).
11. Konkle, T. & Caramazza, A. Tripartite organization of the ventral stream by animacy and object size. *Journal of Neuroscience* **33**, 10235–10242 (2013).
12. Bao, P., She, L., McGill, M. & Tsao, D. A map of object space in primate inferotemporal cortex. *Nature* **583**, 103–108 (2020).